# YOUR DISCRIMINATIVE MODEL IS SECRETLY A GENERATIVE MODEL

## ABSTRACT

Although discriminative and generative models are fundamentally equivalent in understanding data distributions, bridging these paradigms – especially transforming off-the-shelf discriminative models into generative ones – remains challenging. In this paper, we introduce a universal framework that unlocks the generative potential of any discriminative model by directly leveraging the data manifold encoded in its parameter space. Drawing inspiration from the score function used in diffusion models, which measures the distance between a sample and the data manifold in probability space, we generalize this concept to the functional domain. To achieve this, we introduce the Discriminative Score Function (DSF), which quantifies the functional distance between a sample and the data manifold by mapping both into a shared functional space using the Loss Tangent Kernel (LTK), a variant of the Neural Tangent Kernel. Our framework is architecture- and algorithm-agnostic, as evidenced by various architectures such as ViT, ResNet, and DETR on tasks including object detection, classification, and self-supervised learning (*e.g.*, CLIP and DINO). Additionally, our approach extends to applications in image editing, inpainting, and explainable AI(XAI). Finally, we demonstrate the promising potential of DSF by adopting diffusion model techniques for enhanced generation quality.

## 1 INTRODUCTION

Although discriminative and generative models are trained with different objectives, both fundamentally require understanding the distribution of a given dataset. Generative models learn this distribution explicitly as their training objective. Discriminative models, though trained for various objectives, must implicitly capture the underlying data distribution, as effective performance on any task requires understanding the manifold where the data resides. A natural question then arises: can the two switch roles? There has been extensive research on employing generative models for discriminative tasks (Li et al., 2023; Kingma et al., 2014). In contrast, utilizing discriminative models for generation remains largely unexplored and limited.

Discriminative models learn data distribution only implicitly. The models are trained with task-specific objective functions, and not with explicit distribution-matching objectives as in generative models. Consequently, their understanding of data distribution is not direct and is entangled with the training objectives. Prior works necessarily incorporate these loss functions to enable generation from the implied distribution (Haim et al., 2022; Lee et al., 2024). From here, two fundamental limitations arise. First, they cannot serve as a universal framework. They can only be applied to classification models with explicit target labels. The methods fails for object detection, self-supervised models, and other complex objectives where explicit labels are absent. Second, the methods are restricted to label-conditioned generations, whereas true generative ability should be capable of sampling from unconditioned distribution. The works even lack rigorous justification for why their methods should enable generation. Utilizing the data distribution that discriminative models have learned still remains a problem to be solved.

In this paper, we propose the Discriminative Score Function (DSF), a principled framework that transforms arbitrary discriminative models into true generative models. Unlike existing methods, DSF's requirements align perfectly with what discriminative models already possess: the ability to evaluate unseen in-distribution samples. Similar to score functions in diffusion models, DSF pro-

vides gradients that warp generative candidates from noise toward the functional manifold, thereby reducing the functional distance between generated samples and the model's learned representations. Through extensive experiments, we validate that DSF acts as a score function. It universally converts any discriminative model into a generative one and tackles various generative downstream tasks. Our approach is the first to leverage off-the-shelf discriminative models without modifications. This bridges the gap between discriminative and generative paradigms by revealing how models interpret the data manifold.

Specifically, we extend score-matching (Ho et al., 2020b; Song & Ermon, 2019) from probability spaces to functional spaces. We leverage the Loss Tangent Kernel (LTK) (Chen et al., 2023), a variant of NTK (Jacot et al., 2018), which establishes a one-to-one mapping between data and trained models. This allows us to reformulate the generation problem: instead of minimizing distance between generative candidates and the data manifold directly, we map both to functional space via LTK. In this space, the data manifold can be replaced by the pretrained model, transforming our objective into minimizing functional distance between LTK-mapped candidates and the pretrained model. We introduce a surrogate loss to estimate this distance, making DSF applicable to arbitrary off-the-shelf models without modification. To verify DSF's universal versatility, we conduct experiments across multiple training objectives (classification, object detection, self-supervised learning) using diverse architectures (ViT (Dosovitskiy et al., 2021), ResNet (He et al., 2015), DETR (Carion et al., 2020)). Furthermore, DSF enables classical applications such as image editing and inpainting, while also serving as an explainable AI tool that reveals biases and feature entanglements through generated samples. Finally, we demonstrate the potential of DSF by adopting diffusion-inspired sampling strategies. We show that naive adoption of diffusion sampling techniques yields significant performance improvements. By incorporating step-size scheduling and progressive noise injection, we achieve dramatic improvements in generation fidelity. Thereby, our paper not only empirically validates the theoretical equivalence between discriminative and generative paradigms but also suggests promising directions for further refinement.

**Position of our paper.** **(1) Motivation**: We aim to visualize a model's understanding of data by repurposing any task-oriented discriminative model for generative purposes. **(2) Novelty**: Existing methods only work in classification tasks, requiring explicit class labels, and cannot sample directly from a data manifold. However, DSF operates directly in functional space without class label constraints. This allows DSF to measure the distance between sampled images and the data manifold without label dependency, enabling truly unconditional generation from any off-the-shelf discriminative model. **(3) Influence**: Beyond demonstrating theoretical equivalence between discriminative and generative paradigms, DSF opens new possibilities for model understanding and deployment. It transforms any pretrained discriminative model into a generative model without additional training, and provides a powerful tool for revealing learned biases and representations through generation.

## 2 RELATED WORKS

**Score-based generative models** generate samples by approximating the probability distribution of the target data (Hyvärinen & Dayan, 2005; Song et al., 2020b). Since direct approximation is infeasible, a diffusion process is used, where noise is added to the target distribution to transform it into a known distribution such as Gaussian (Ho et al., 2020b; Song et al., 2020a) and then reversed to recover the target. In essence, these models iteratively transition from a Gaussian distribution to the data manifold. Training such models requires delicate methodologies, including learning a score function instead of modeling the full distribution (Ho et al., 2020a) and handling data differently at each time step. Noisy classification is also necessary to perform different classifications at different time steps for class-conditional generation (Dhariwal & Nichol, 2021).

**Image Generation using classification models** has branched into two directions: 1. Gradient-Ascent-Based Approaches and 2. Maximal-Margin-Based Techniques. The former approaches are commonly used in feature visualization such as DeepDream (Mordvintsev et al., 2015; Olah et al., 2017) and energy-based models. Fundamentally, they function like adversarial attacks, optimizing an image to activate a specific class or neuron. However, naive gradient ascent often fails to produce meaningful results, necessitating the use of additional conditions or objectives, such as requirement of specific architecture (Yin et al., 2020), frequency-based regularization (Fel et al., 2023), total vari-

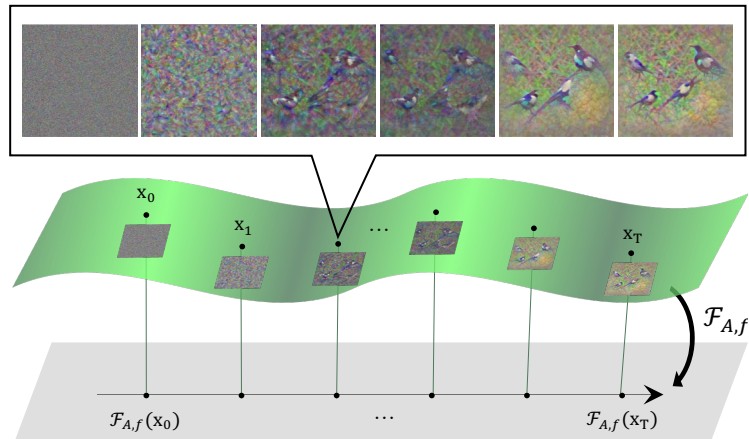

Figure 1: Conceptual diagram of our process. we update image $x_t$ by utilizing DSF in the functional space with Functor $\mathcal{F}_{\mathcal{A},f}$.

ation (Mahendran & Vedaldi, 2015), or pre-trained generative models (Grathwohl et al., 2019). In energy-based models, researchers frequently add auxiliary losses to fine-tune the classifier, improving the interpretability of gradient ascent results (Yang et al., 2021). Alternatively, some methods (Santurkar et al., 2019) jointly optimize classification and image generation objectives. On the other hand, the latter (Maximal-Margin-based) methods leverage the stationarity condition and a theoretical link between Support Vector Machines and deep-learning models to reconstruct or generate training data (Lyu & Li, 2020; Haim et al., 2022; Buzaglo et al., 2023). Typically, they also operate under highly overfitted settings with simple architectures (e.g., basic ConvNets or MLPs) and small datasets (e.g., MNIST, CIFAR10) (Goodfellow et al., 2016). Although there exists a line of work (Lee et al., 2024) which scales to higher-resolution datasets, it remains confined to classification models.

## 3    PRELIMINARIES

**Kernel Methods**   are conventional machine learning techniques where a bivariate kernel $k(\cdot, \cdot)$ represents the inner product in a high-dimensional feature space, i.e., $k(x, y) = \langle \phi(x), \phi(y) \rangle$. Reproducing Kernel Hilbert Space (RKHS) guarantees the existence of the space $\phi$ where the closeness of $\phi(x)$ and $\phi(y)$ implies the closeness of $x$ and $y$. Fixing one input in the kernel, we obtain a univariate kernel $k(\cdot, x)$. It is worth noting that this kernel exists in a functional space, i.e., it is a function.

**Neural Tangent Kernel**   tries to explain the generalization ability of deep learning models using conventional kernel methods. It suggests that for every deep learning network being trained, a corresponding kernel exists. NTK $\Theta(x, x')$ is defined by the gradients of the given deep learning model, $\Theta(x, x') = \nabla_\theta f(x)^\top \nabla_\theta f(x')$, reflecting how the model trains in the input space. However, as NTK does not account for loss dynamics, Loss Tangent Kernel (LTK) Chen et al. (2023) extends NTK by incorporating the derivative of the loss function. The LTK is expressed as: $K(x, x') = \nabla_f \ell(f(x; \theta))^\top \Theta(x, x') \nabla_f \ell(f(x'; \theta))$.

## 4    DISCRIMINATIVE SCORE FUNCTION

In this section, we introduce our Discriminative Score Function (DSF) through a step-by-step derivation. First, we generalize the score-based model framework in Sec. 4.1 and establish that score models fundamentally minimize the distance between generated images and the data manifold. Then we leverage key insight: discriminative models inherently demonstrates their understanding of the data manifold. In Sec. 4.2, we introduce the Functor concept, which explicitly maps models to their learned manifolds using the Loss Tangent Kernel (LTK). Finally, in Sec. 4.3, we synthesize these

components to define DSF, a metric that measures the distance between images and the manifold through functional space.

### 4.1 CONNECTION BETWEEN SCORE-BASED MODEL AND GEOMETRY

The goal of generative model is estimating data distribution $p(x)$ using a model $f$. There are two important points to consider. First, $p(x)$ is unknown and what we do know is the sampled dataset $\mathcal{D}_s \sim p(x)$. Second, the image space $\mathbb{R}^{hw}$ is so high-dimensional that the manifold $\mathcal{M} \subseteq \mathbb{R}^{hw}$ where actual data resides is relatively small and hard to approximate.

The direct way to solve this issue is to first sample $x_0$ from a known distribution and iteratively update it for $T$ steps to generate $x_T$ toward the manifold $\mathcal{M}$. Since we have no access to $\mathcal{M}$, the above process should be done by an approximation using $\mathcal{D}_s \subseteq \mathcal{M}$. This can be written as the following:

$$
\begin{aligned}
x_{t+1} &= x_t - \eta_t \nabla_x d(x_t, \mathcal{M}) := x_t - \eta_t \nabla_x \int_{\mathcal{M}} d(x_t, x)dx \\
&\simeq x_t - \eta_t \nabla_x \sum_i d(x_t, \zeta_i),
\end{aligned}
\tag{1}
$$

where, $\zeta_i$ is the $i$th sample of $\mathcal{D}_s$ and $\eta_t$ is the step size.

Direct implementation of Eq. (1) is almost impossible for two reasons: 1. It is difficult to find a metric $d(\cdot, \cdot)$ that can well-represent the manifold, $\mathcal{M}$, since the manifold takes up only a small portion in the high-dimensional space and has an uneven shape. 2. The amount of calculation is proportional to the number of samples in our data $\mathcal{D}_s$, *i.e.*, proportional to $|\mathcal{D}_s|$.

Score-based models resolves the implementation problem by translating Eq. (1) from the geometric framework $\mathcal{M}$ to a probability space. The resulting equation, now framed within the probability space, is provided below:

$$
x_{t+1} = x_t + \eta_t \nabla_x \log p(x_t).
\tag{2}
$$

Here, the term score function refers to the gradient of the log probability density function, $\nabla_x \log p(x_t)$. Replacing the score function in Eq. (2) with $-\nabla_x d(x_t, \mathcal{M})$ brings us back to Eq. (1). The detailed derivation of the connection between the score function and the manifold distance term $d(x_t, \mathcal{M})$ can be found in Appendix. B.

As we only have a finite dataset, the distance $d$ in Eq. 1 should satisfy the following condition:

$$
d(\cdot, \mathcal{D}_s) \simeq d(\cdot, \mathcal{D}_t), \quad \forall \mathcal{D}_t \subseteq \mathcal{M},
\tag{3}
$$

where $\mathcal{D}_s$ and $\mathcal{D}_t$ represent the training dataset and any test dataset drawn from the manifold $\mathcal{M}$, respectively. Intuitively, this means the metric should generalize the manifold effectively even when samples are sparse. Consequently, any subset from the manifold $\mathcal{M}$ can provide a reliable estimate for $d(\cdot, \mathcal{M})$.

Interestingly, discriminative models inherently satisfy this condition. In discriminative settings, achieving consistent performance from training $\mathcal{D}_s$ to test sets $\mathcal{D}_t$ precisely corresponds to generalization. This is exactly the property we want our distance metric to possess. Thus, the central question now becomes: How can we explicitly leverage this generalization ability?

### 4.2 MAP THE MANIFOLD TO FUNCTION SPACE

To directly utilize the discriminative model, we need to map the data manifold to the function space. The key insight is that the training procedure itself serves as this mapping: datasets from the same manifold should produce similar trained models.

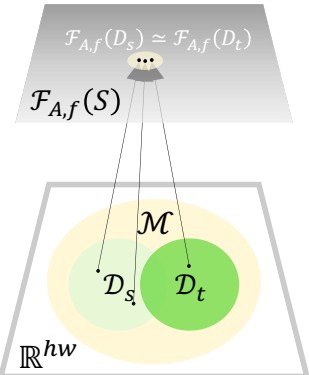

Figure 2: Structure of Functor $\mathcal{F}_{A,f}$. $\mathcal{D}_s$ and $\mathcal{D}_t$ sampled from the manifold is mapped from $\mathcal{R}^{hw}$ space to $\mathcal{F}_{A,f}(S)$, $\forall S \subseteq \mathbb{R}^{hw}$ space. The two are mapped to similar space due to the generalization property of a discriminative model.

We formalize this with a Functor $\mathcal{F}$ that maps a training setup to a trained model $f^\star$:

$$\mathcal{F}: \quad (\mathcal{D} \times \mathcal{A} \times f) \rightarrow f^\star, \tag{4}$$

where $\mathcal{D}$ is the dataset, $\mathcal{A}$ is the training algorithm (*e.g.*, SGD), and $f$ is the network architecture.

Fixing $\mathcal{A}$ and $f$ (*e.g.*, ResNet-50 and SGD), we obtain $\mathcal{F}_{\mathcal{A},f}: \mathcal{D} \rightarrow f^\star$. This Functor can map any data—including generated candidates—to the function space, enabling direct use of the discriminative model. (See Fig. 2.)

Since $\mathcal{F}_{\mathcal{A},f}(\mathcal{D}_s)$ is the trained discriminative model itself and generalizes to any test set from $\mathcal{M}$, we can replace the manifold distance in Eq. (1) to obtain an iterative generation process illustrated in Fig. 1:

$$\begin{aligned}
x_{t+1} &= x_t - \eta_t \nabla_x d(\mathcal{F}_{\mathcal{A},f}(x), \mathcal{F}_{\mathcal{A},f}(\mathcal{M})) \\
&\simeq x_t - \eta_t \nabla_x d(\mathcal{F}_{\mathcal{A},f}(x), \mathcal{F}_{\mathcal{A},f}(\mathcal{D}_s)).
\end{aligned} \tag{5}$$

This substitution is valid because $\mathcal{F}_{\mathcal{A},f}$ is locally continuous—small changes in the dataset produce small changes in the trained model (Bousquet & Elisseeff, 2002; Elisseeff et al., 2005; Xu & Mannor, 2012; Hardt et al., 2016). This framework directly leverages the discriminative model since $\mathcal{F}_{\mathcal{A},f}(\mathcal{D}_s)$ itself is the trained discriminative model. What remains is to derive a good functor $\mathcal{F}$.

### 4.3 DIRECT USE OF DISCRIMINATIVE MODEL THROUGH LTK

**Loss Tangent Kernel as a Functor** To directly compute the Functor $\mathcal{F}$ without explicit training, we employ the Loss Tangent Kernel (LTK) (Chen et al., 2023). LTK provides a training-free equivalent to the trained model by integrating loss derivatives with the Neural Tangent Kernel (NTK).

The LTK $K$ measures the similarity between data points through their loss gradients:

$$K(x,x') \triangleq \langle k(\cdot,x), k(\cdot,x') \rangle = \nabla_f \ell(f(x))^\top \Theta \nabla_f \ell(f(x')) = \nabla_\theta \ell(f(x))^\top \nabla_\theta \ell(f(x'))$$

$$\nabla_f \ell(f(x)) \in \mathbb{R}^{C \times 1},\ \nabla_\theta \ell(f(x)) \in \mathbb{R}^{P \times 1},\ \Theta \in \mathbb{R}^{C \times C}, \tag{6}$$

where $C$ is the number of classes, $P$ is the number of parameters and $\Theta$ is the NTK.

Replacing $\mathcal{F}_{\mathcal{A},f}(x)$ with $k(\cdot, x)$ in Eq. (5):

$$x_{t+1} = x_t - \eta_t \nabla_{x_t} d(k(\cdot, x_t), k(\cdot, \mathcal{D}_s)) \tag{7}$$

$$\simeq x_t + \eta_t \nabla_{x_t} K(x_t, \mathcal{D}_s) \tag{8}$$

$$= x_t + \frac{\eta_t}{|\mathcal{D}_s|} \nabla_{x_t} \{ \nabla_\theta \ell(f(x_t;\theta))^T \sum_i^{|\mathcal{D}_s|} \nabla_\theta \ell(f(\zeta_i;\theta)) \} \tag{9}$$

$$\simeq x_t - \eta_t \nabla_{x_t} d(\nabla_\theta \ell(f(x_t;\theta)), -\theta) \tag{10}$$

The derivation follows three steps: (1 - eq. (7)) replacing $\mathcal{F}$ with the kernel representation $k(\cdot, x)$ in the functional space; (2 - eq. (8)) approximating distance with negative inner product in RKHS; (3 - eq. (10)) leveraging the maximal margin property where loss gradients align with parameters at convergence (Ji & Telgarsky, 2020; Yun et al., 2021), allowing us to approximate $\sum_i \nabla_\theta \ell(f(\zeta_i;\theta))$ with $-\theta$. Note that while the Functor formally maps datasets to models, we extend this notation to individual samples for clarity. In practice, we process batches, effectively applying the framework to distributions.

**Label-Free Loss via Training Dynamics Matching** Direct use of label-dependent losses prevents unconditional generation and may be unavailable in practice. We address this by introducing a surrogate loss that exhibits similar training dynamics.

We select augmentation invariance loss as it naturally decreases during training, mirroring label-dependent loss behavior. This is supported by evidence that augmentation invariance emerges as models improve generalization (Tarvainen & Valpola, 2017; Laine & Aila, 2016; Sohn et al., 2020).

More specifically, we replace $\ell(f(x_t;\theta))$ in Eq. (10) with:

$$\ell_{aug}(x_t;\theta) = \frac{1}{2}\|f(x_t;\theta_{sg}) - f(\mathcal{A}(x_t);\theta)\|^2, \tag{11}$$

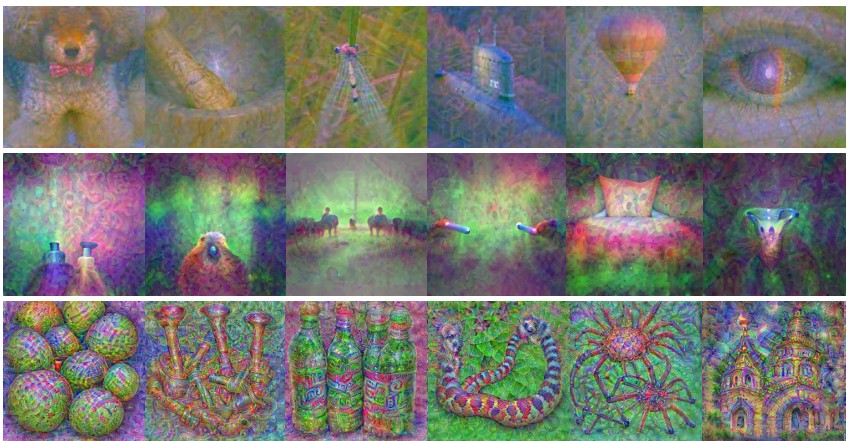

Figure 3: Examples of **unconditionally generated images** using various discriminative models. Top: DINOv2/LVD-124M (self-supervised learning). Middle: DeTR/COCO (object detection). Bottom: ResNet/Imagenet (image classification).

where $\mathcal{A}(\cdot)$ is an augmentation operation (*e.g.*, flipping) and $sg$ denotes stop gradient. This enables unconditional generation while maintaining the essential dynamics of the original loss, resulting in

$$x_{t+1} = x_t - \eta_t \nabla_x d(\nabla_\theta f(x_t; \theta)\lambda, -\theta), \tag{12}$$

where $\lambda \triangleq f(x_t; \theta) - f(\mathcal{A}(x_t); \theta) \in \mathbb{R}^{C \times 1}$ and $\nabla_\theta f(x_t; \theta) \in \mathbb{R}^{P \times C}$.

In eq. (12), $-\nabla_x d(\nabla_\theta f(x_t; \theta)\lambda, -\theta)$ acts as a score function and we define it as a Discriminative Score Function (DSF). Note that DSF can be obtained solely from a pretrained network. In our experiment, we set $d(a, b) = \|a/\|a\|_2 - b/\|b\|_2\|$.

## 5 EXPERIMENTS

**Implementation Details** In this paper, we utilized ResNet (He et al., 2015), DINOv2 (Oquab et al., 2024), and DeTR (Carion et al., 2020) models as the backbone for direct generation of $256 \times 256$ images. All models were obtained from the official repositories. DINOv2 is based on ViT (Vision Transformer) (Dosovitskiy et al., 2021) architecture and was trained on LVD-142M dataset. DeTR was trained using COCO dataset (Lin et al., 2014). For conditioning, we employed both ResNet and OpenCLIP (Cherti et al., 2023) models. We used OpenCLIP as a base model. During the optimization process, we set gradient clipping value to 1e-5 and fixed step size $\eta_t$ to 2 for all $t$ across all models. We calculate the weighted sum of the score term (DSF) and the conditional term for conditional generations. For ResNet the weight of the score term is set to 10, and the conditional term to 20. We also applied variance norm as a regularizer (Mahendran & Vedaldi, 2015). The regularizer is added to Eq. (13) with its weight set to 20. Since the size of DSF is proportional to the number of parameters, we adjusted the hyperparameters for the remaining experiments to match the size of the DSF.

### 5.1 UNCONDITIONAL GENERATION

What we obtain through Eq. (12) is the distance between an arbitrary point $x \in \mathbb{R}^{HW}$ and the image manifold $\mathcal{M}$. A question naturally arises. **Is naively applying DSF to the generation process enough to generate images unconditionally from a discriminative model?** Can we approximate the distribution $p(x)$ of datasets, the fundamental objective of generative models?

Extensive experiments were performed to empirically answer the question. We conducted the demonstration as fairly as possible with different datasets, architectures and algorithms. For datasets, ImageNet, LVD-124M, and COCO dataset were chosen. We selected three models of different architectures and algorithms: ResNet for ImageNet classification, DeTR for COCO object detection, and DINOv2 for self-supervised learning. We only utilized famous models from public repositories

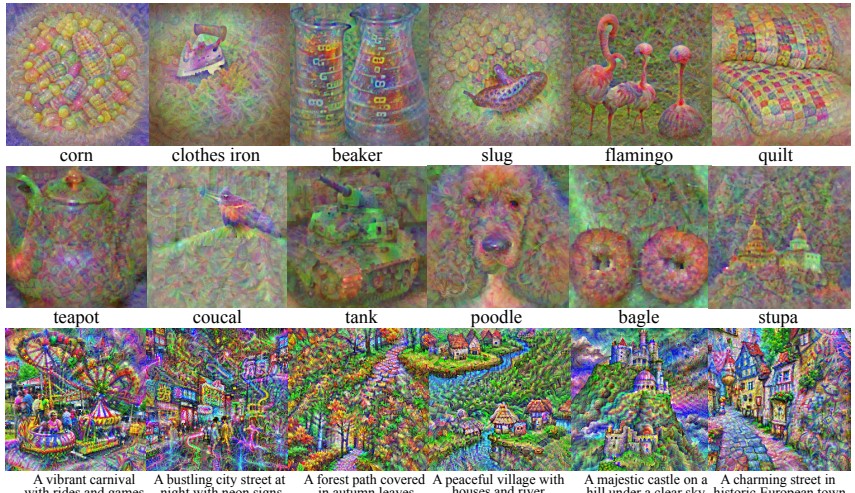

Figure 4: Examples of **conditionally generated images** using DSF. Top: Both the score model and the conditional model used ResNet/Imagenet. Middle: The score model used DINOv2/LVD-124M, and the conditional model used ResNet/Imagenet. Bottom: The score model used ResNet/Imagenet, and the conditional model used CLIP.

for transparency. No post-training was done because we are to show that generation is possible without modification or tuning for any arbitrary model.

For datasets, DSF successfully covered datasets of various aspects. DSF successfully generated images under ImageNet setting, which is considered the most common and general one at the moment. The high-quality image output using LVD-124M, which is one of the largest datasets used for vision tasks, proves that the generation succeeds regardless of the size of the training dataset. Tests on COCO dataset that includes images of various objects, shapes and sizes in high-resolution imply that DSF can even handle high-resolution datasets. The results are also favorable for different architectures and algorithms. Fig. 3 shows the result of unconditional generation using three pairs of different models and algorithms. All three pairs successfully generated images of realistic shapes.

Fig.1 illustrates the conceptual generation process of DSF. Like diffusion models, images are generated through an iterative process with respect to a score function (DSF) that starts from noise. The difference from the conventional diffusion model lies in the space: the conventional score function measures the probability manifold (Ho et al., 2020b), while DSF measures the functional manifold.

## 5.2 CONDITIONAL GENERATION

Since the diffusion model can generate images conditionally, DSF should also be able to do so. To test this capability, we implemented a naive version following guided diffusion (Dhariwal & Nichol, 2021). Specifically, we added a conditional loss term to Eq. 12 as follows:

$$x_{t+1} = x_t \underbrace{-\nabla_x d(\nabla_\theta f(x_t; \theta)\lambda, -\theta)}_{\text{score}} + \underbrace{\nabla_x \log p_\phi(y|x_t)}_{\text{condition}}. \tag{13}$$

Here, $p_\phi$ refers to any guidance model, which ensures guidance during the generation process. By simply summing the DSF and the guidance, we can generate conditional images accordingly.

In Fig. 4 we used common pretrained classification model in a plug-and-play manner for generation. Our score and condition model for DSF successfully replaced the ones in the diffusion equation and accomplished image generations. Furthermore, the last row of Fig. 4 suggests that not only classifier guidance, but also text-to-image (T2I) with a specific prompt input is possible when using CLIP.

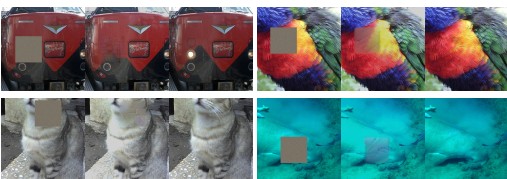

Figure 5: Examples of **editing** through DSF. The top row contains original images while the bottom row shows edited images.

Figure 6: Examples of the **inpainting** task. For each set, the left is the inpainted image, the middle is the reconstructed one, and the right is the original image.

## 6 APPLICATIONS

### 6.1 IMAGE EDITING

What a diffusion model learns is the reverse process of converting an image to pure noise. Based on this property, editing tasks in diffusion include sending the object image to a noise-like image and retrieving it back. The condition is injected into this process in the form of masking (Couairon et al., 2022). For the case of DSF, what it conducts is not the denoising process but the direct estimation of manifold distribution $p(x)$. Editing is therefore possible at image level without adding noise to the target image. Masking is also unnecessary. The result of directly applying conditional generation to real image without adding noise nor masking is shown in Fig. 5. The outstanding point is that the semantic has changed without largely deforming the input images. For example, in the photo of a chimpanzee, a spot of light fallen at the background is naturally transformed to a head of iguana. The body of the animal is trivially deformed to consist shadow and body of the iguana. This was possible by consistently injecting our score function as restraint, preventing the result from straying away from the manifold.

### 6.2 IMAGE INPAINTING

Image inpainting is a technique that visually restore the damaged or omitted part of the target image. To enable this, the ability to generate new contents according to the whole context it understood is mainly required. Diffusion model is suitable for inpainting tasks in this sense. Similarly, DSF can serve as an effective inpainting model. To evaluate its performance, we masked random regions of an image by generating empty patches. Although DSF is calculated to the entire image, updates were restricted to the masked areas. To guide the restoration process, we introduced a simple regularization condition that enforces consistency at the boundaries of the masked regions. With this straightforward setup and the naive application of the DSF, we successfully restored the missing regions, as shown in Fig. 6.

### 6.3 GLOBAL EXPLANATION

Our methodology centers on proposing a distance measure using a conditioned functor $\mathcal{F}_{\mathcal{A},f}$ with fixed $\mathcal{A}$ and model $f$, where DSF moves $x$ to the center of manifold $\mathcal{M}$ through the lens of model $f^\star$ learned from $\mathcal{M}$. This approach serves as an explainable AI (XAI) tool for feature visualization that goes beyond conventional gradient descent methods by offering mathematical justification for how generated images reflect the underlying manifold—unlike adversarial attacks which, despite using gradient ascent to update images,

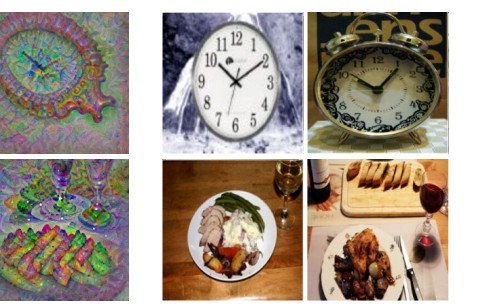

Figure 7: Comparison between ours and the existing feature visualization method: (Left) DSF, (Middle) Real ImageNet images of the targeted class (Right) Visualizations produced by (Fel et al., 2023).

provide no insights into actual decision criteria. By generating complete images that reflect the manifold, DSF effectively identifies subtle biases and entanglements in data: for instance, when applied to a ResNet model trained on ImageNet, DSF consistently generated "analog clock" images showing 10:10 (reflecting advertising bias in the training data) and "plate" images that included contextual elements like wine glasses and dishes, revealing data entanglements that other methods like (Fel et al., 2023) fail to preserve. These results demonstrate DSF's unique capability to reveal and visualize biases and entanglements while maintaining mathematical rigor, offering valuable insights into dataset characteristics.

## 6.4 UNLOCKING THE POTENTIAL OF DSF THROUGH DIFFUSION MODEL

Since DSF is derived from generalizing score-based models, we can naturally adopt techniques from diffusion models to enhance generation quality. While our primary goal was to demonstrate the feasibility on converting discriminative model into generative ones, we adopt simple strategy: fixed step sizes in our base experiments. However, diffusion models usually achieve superior results through adaptive step-size scheduling and intermediate noise injection for bias correction Song et al. (2020a); Ho et al. (2020a). We applied these techniques in their simplest form: linear step-size decay coupled with proportional noise injection. Despite this naive implementation, the results were remarkable.

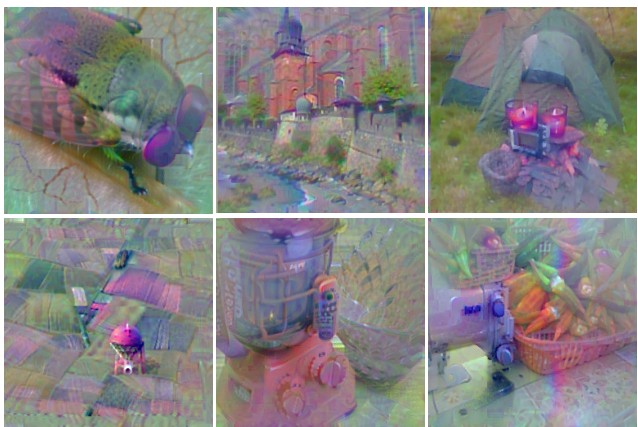

Figure 8: Result of unconditionally generated images using DINOv2 Oquab et al. (2024), for sampling, we adopted linear step scheduling and fixed noise injection.

As shown in Fig. 8, unconditional generation from DINOv2 (Oquab et al., 2024) exhibited dramatically reduced artifacts and significantly enhanced detail compared to our baseline results in Fig. 3. The generated images now display cleaner edges, more coherent structures, and fewer visual distortions. These improvements demonstrate that DSF extends beyond theoretical visualization to practical generation. The successful adoption of diffusion techniques not only unlocks DSF's full potential but also empirically strengthens its connection to score-based models. This suggests that the wealth of techniques developed for diffusion models could be systematically applied to DSF, opening promising avenues for future refinement.

## 7 CONCLUSION AND DISCUSSION

In this paper, we demonstrate that any discriminative model can serve generative purposes through our proposed Discriminative Score Function (DSF). By interpreting discriminative models as functional measures of the data manifold, DSF enables universal transformation from discrimination to generation without architectural modifications or retraining. We show that DSF successfully handles diverse visual tasks including unconditional generation, image editing, and inpainting across various model architectures and training objectives. Beyond practical applications, DSF serves as a powerful interpretative tool, revealing latent biases and learned representations within discriminative models. Furthermore, by adopting diffusion-inspired techniques such as step-size scheduling and noise injection, we dramatically improve generation fidelity, demonstrating DSF's potential to bridge theoretical insights with practical performance. Looking forward, DSF's reliance on fundamental discriminative properties rather than task-specific objectives suggests broad applicability beyond computer vision. The framework could naturally extend to Natural Language Processing, where language models implicitly capture textual distributions, or Reinforcement Learning, where value functions encode environmental dynamics. These directions, combined with more sophisticated sampling strategies borrowed from diffusion models, present promising avenues for future research in unified discriminative-generative modeling.

**Usage of LLM**  We used Large Language Models Anthropic (2024) to improve the grammatical accuracy and readability of this manuscript. All technical content, ideas, experimental designs, and scientific contributions are entirely our own original work.

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

## A    IMPACT STATEMENT

This paper introduces a schema that transforms any discriminative model into a generative model. By repurposing pre-trained discriminative models in this manner, we can substantially save computational resources otherwise spent on training separate generative models. In addition, the proposed framework offers a way to reveal inherent biases – either from the model itself or from the underlying dataset – by visualizing and inspecting the learned data manifold.

## B    GENERALIZATION OF SCORE MODEL

In this section, we demonstrate how the score function can be understood geometrically. Specifically, consider the following generation process through a score-based model:

$$x_{t+1} = x_t + \lambda_t \nabla \log p(x_t), \tag{14}$$

where $p(x)$ is the probability distribution corresponding to the desired manifold of data. Our goal is to approximate $p(x)$ in a way that captures the underlying structure of this manifold.

A natural way to view $p(x)$ is as a continuous generalization of **Kernel Density Estimation (KDE)**. Recall that in KDE, an empirical density estimate $\hat{p}(x)$ can be written as:

$$\hat{p}(x) = \frac{1}{n} \sum_{i=1}^{n} K(x, x_i). \tag{15}$$

Extending this notion to a continuous integral over the manifold $M$, we can write:

$$p(x) = \int_M K(x, u) \, du. \tag{16}$$

From this perspective, the corresponding **score function** (i.e., the gradient of the log-density) becomes:

$$\nabla_x \log p(x) = \nabla_x \log \int_M K(x, u) \, du. \tag{17}$$

Geometrically, this gradient term can be interpreted as a "pull" of the point $x$ toward regions of high data density on the manifold $M$. In other words, it steers $x$ toward the locations where the data (or the desired manifold) is most concentrated.

Moreover, when viewed purely in terms of geometry, one may simplify the problem by omitting the log for computational convenience or stability. In that case, the **diffusion-like** or **flow-based** perspective leads us to solve the more direct generalization:

$$\nabla_x \int_M d(x, u) \, du, \tag{18}$$

where $d(x, u)$ is some distance or kernel-based measure between $x$ and points $u$ in the manifold $M$. This highlights how the iterative update can be thought of as moving $x$ along the manifold to regions of higher density.

Figure 12 provides a **conceptual diagram** illustrating this process. A notable detail in practical implementations is the **static update step** $\Delta$; because this step size does not adapt in certain methods, it may lead to high-frequency artifacts or other unwanted oscillatory behaviors in the generated samples.

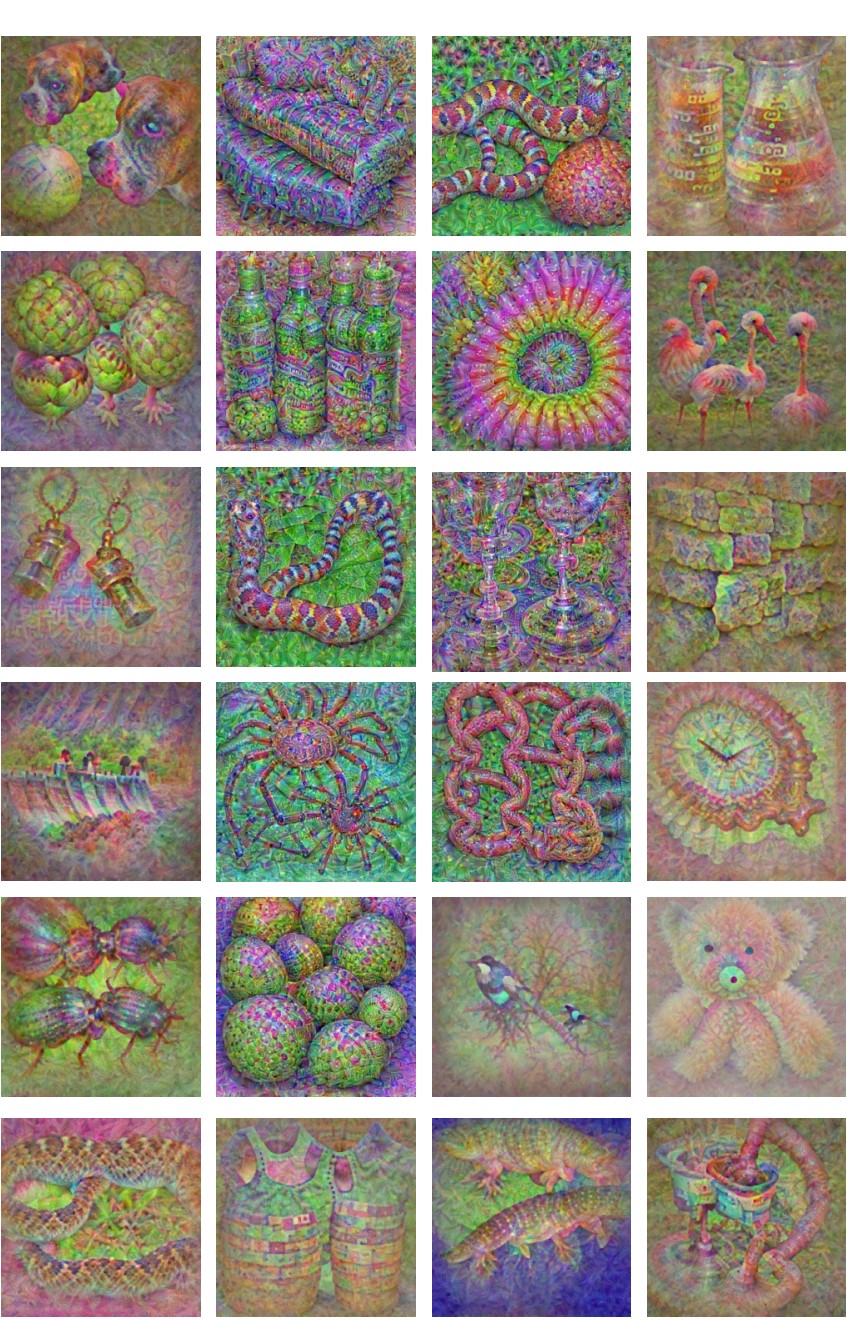

Figure 9: More images generated by ResNet50/ImageNet.

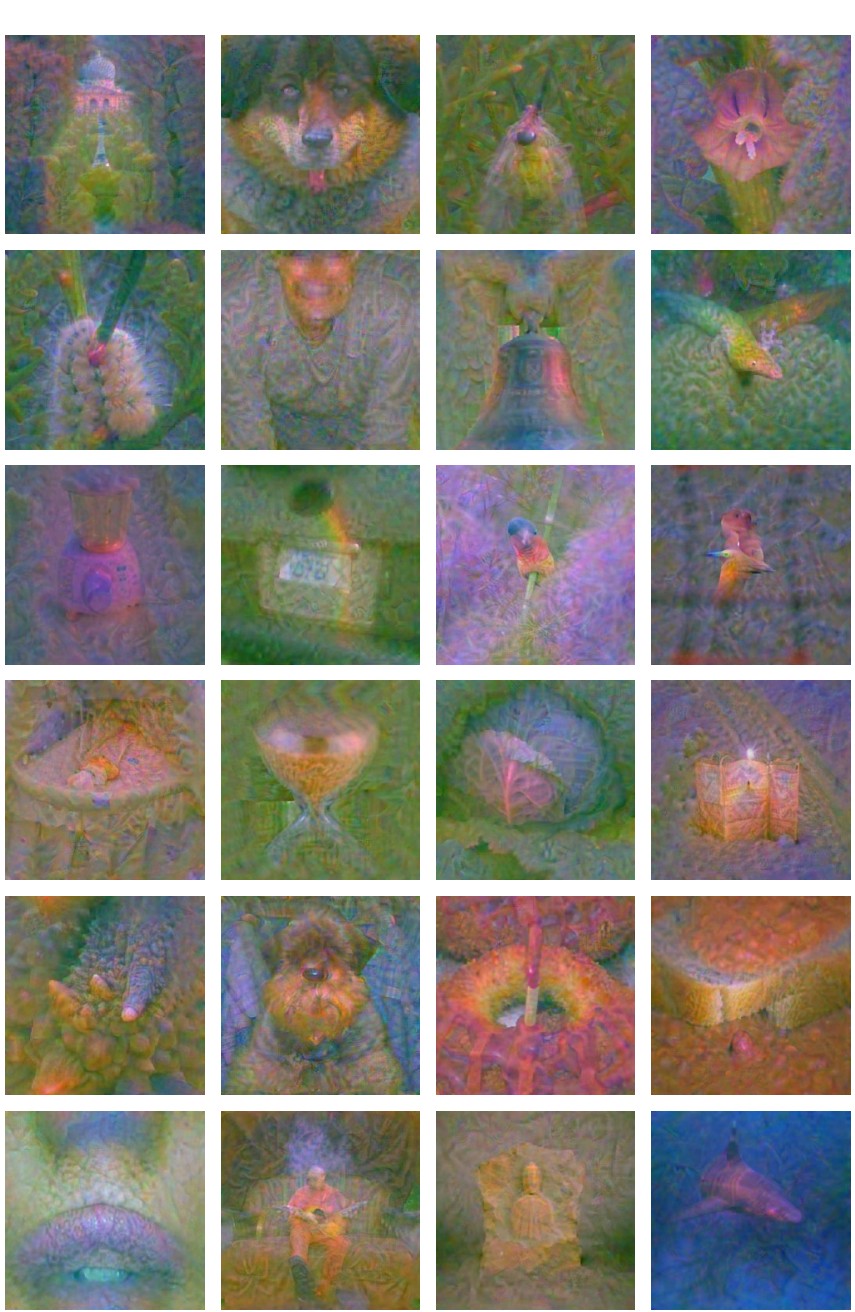

Figure 10: More images generated by DINOv2/LVD-124M.

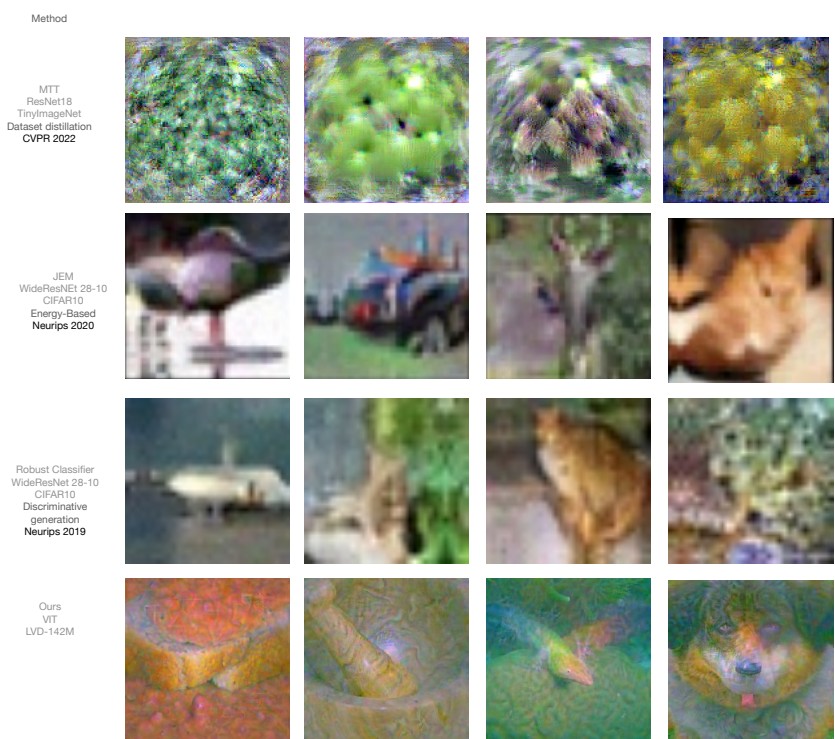

Figure 11: Comparison between other benchmarks and ours.

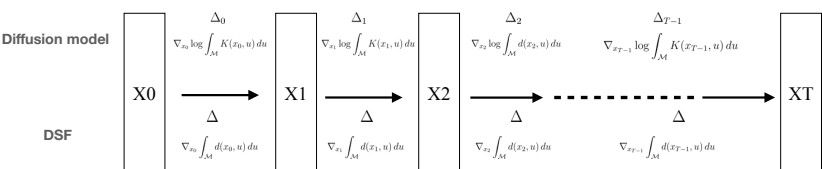

Figure 12: Comparison between diffusion models and DSF. Both methods solve the same procedure. As probability can be viewed as the limiting case of Kernel Density Estimation, it can be written as $\nabla_x \log \int_{\mathcal{M}} K(x, u) du$, where $\Delta$ refers to step size.

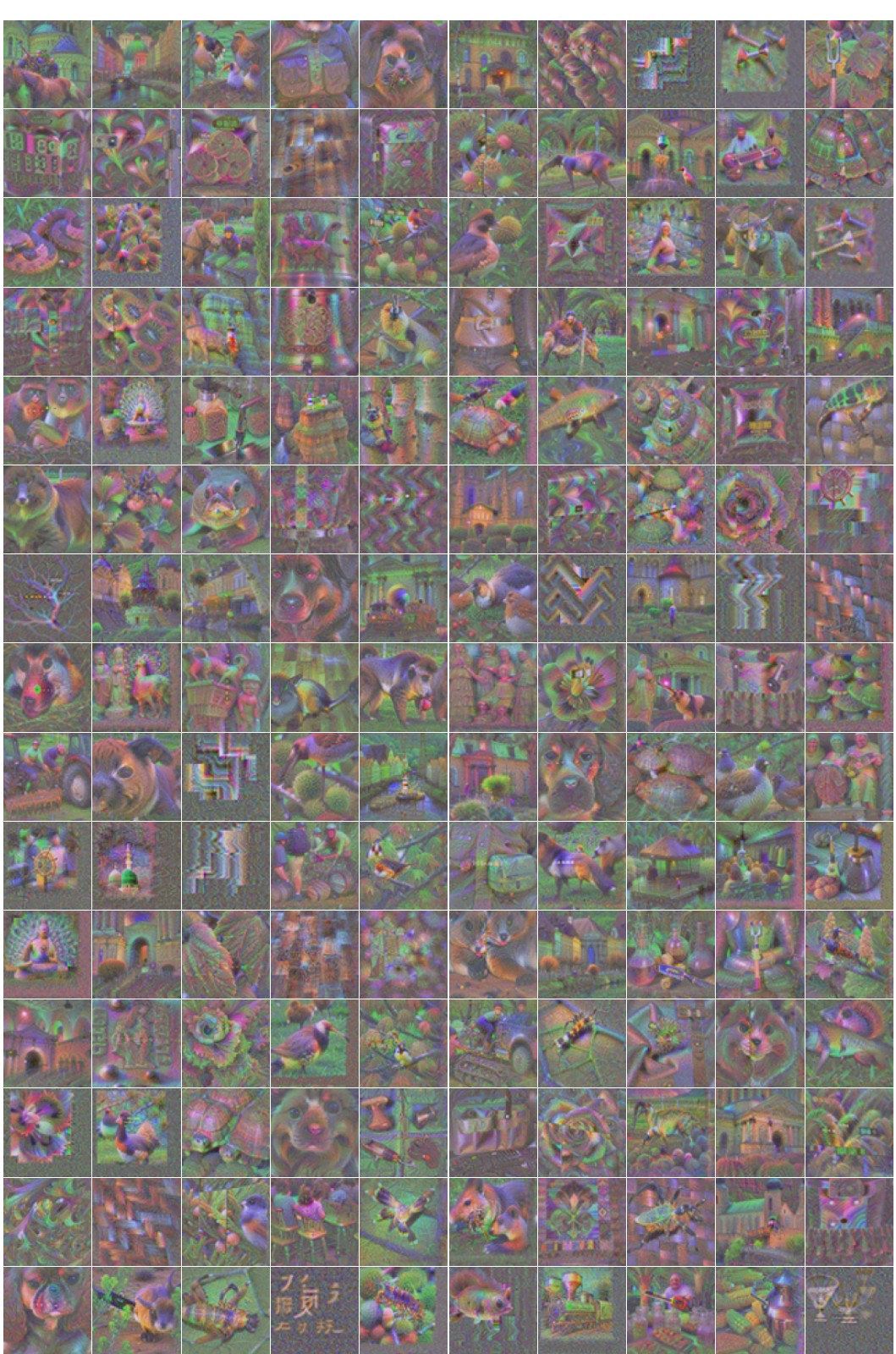

Figure 13: uncurated images of DSF (1)

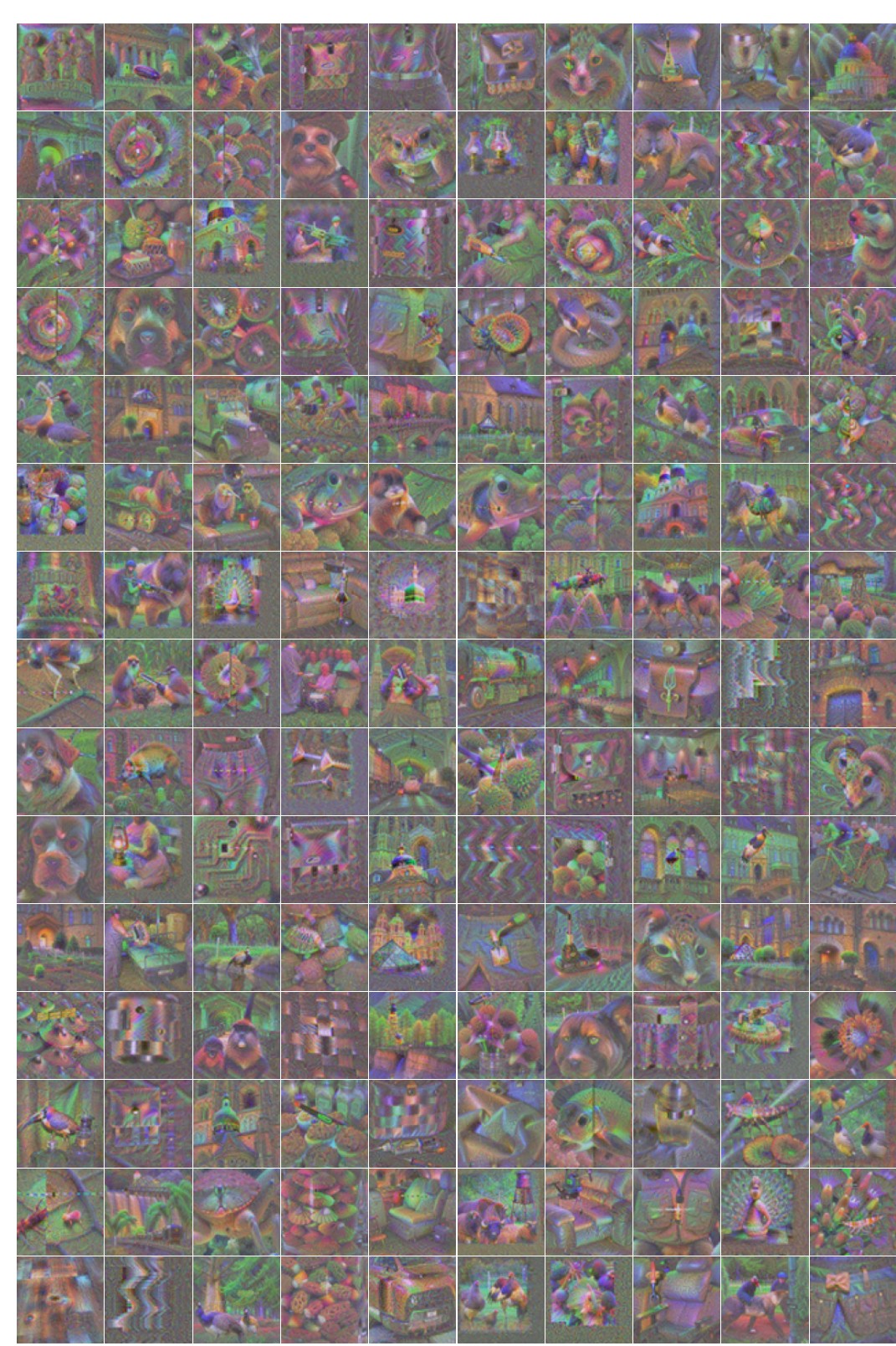

Figure 14: uncurated images of DSF (2)

1026
1027
1028
1029
1030
1031
1032
1033
1034
1035
1036
1037
1038
1039
1040
1041
1042
1043
1044
1045
1046
1047
1048
1049
1050
1051
1052
1053
1054
1055
1056
1057
1058
1059
1060
1061
1062
1063
1064
1065
1066
1067
1068
1069
1070
1071
1072
1073
1074
1075
1076
1077
1078
1079

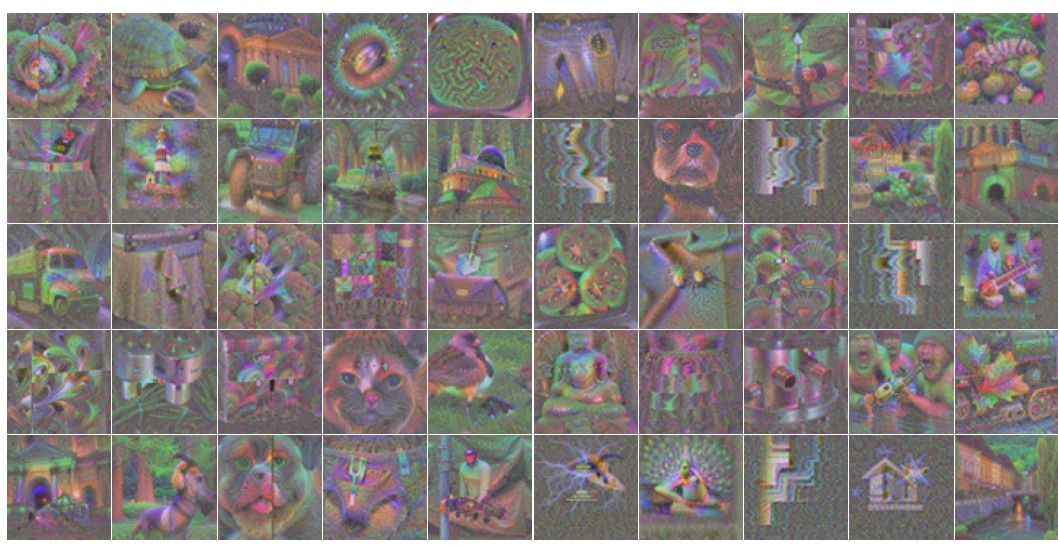

Figure 15: uncurated images of DSF (3)

