

Figure 9: More images generated by ResNet50/ImageNet.

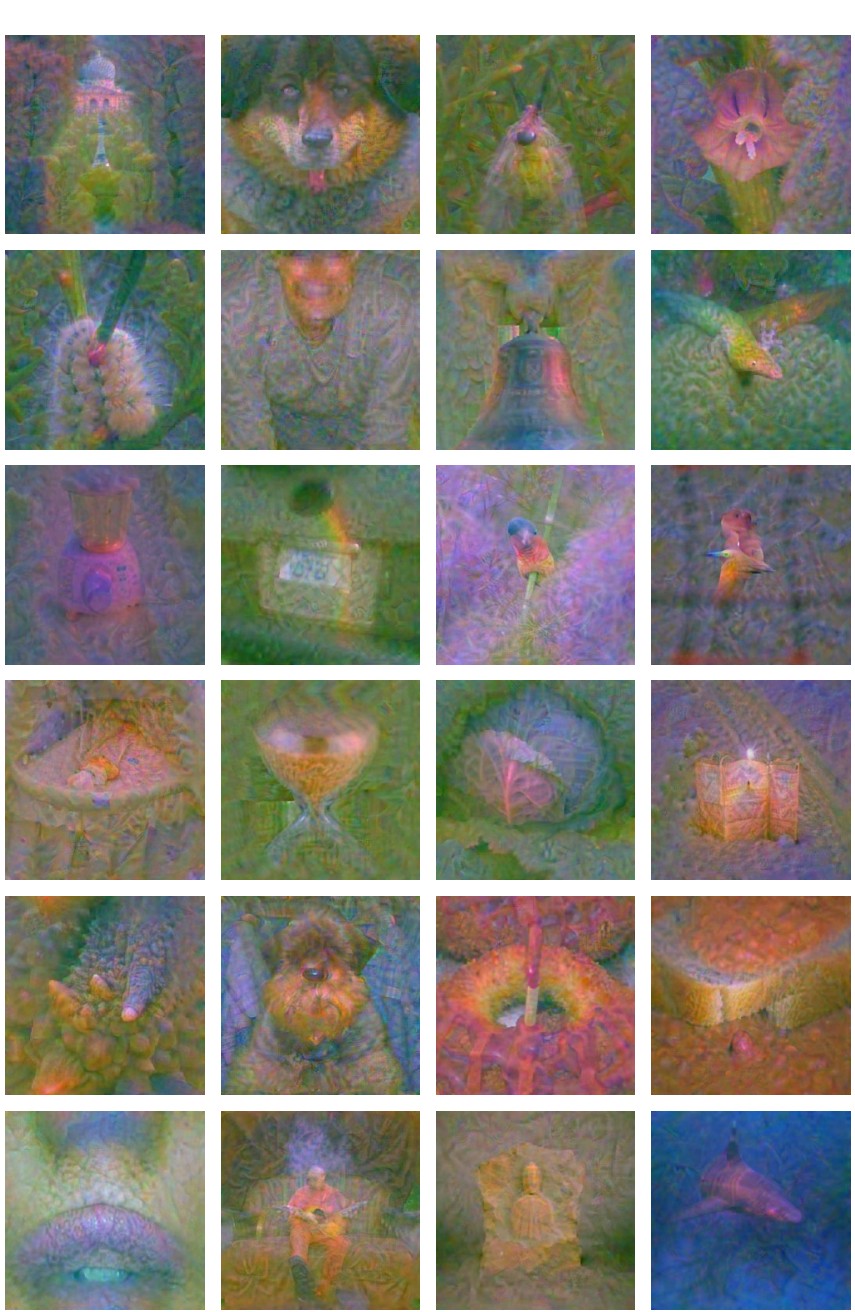

Figure 10: More images generated by DINOv2/LVD-124M.

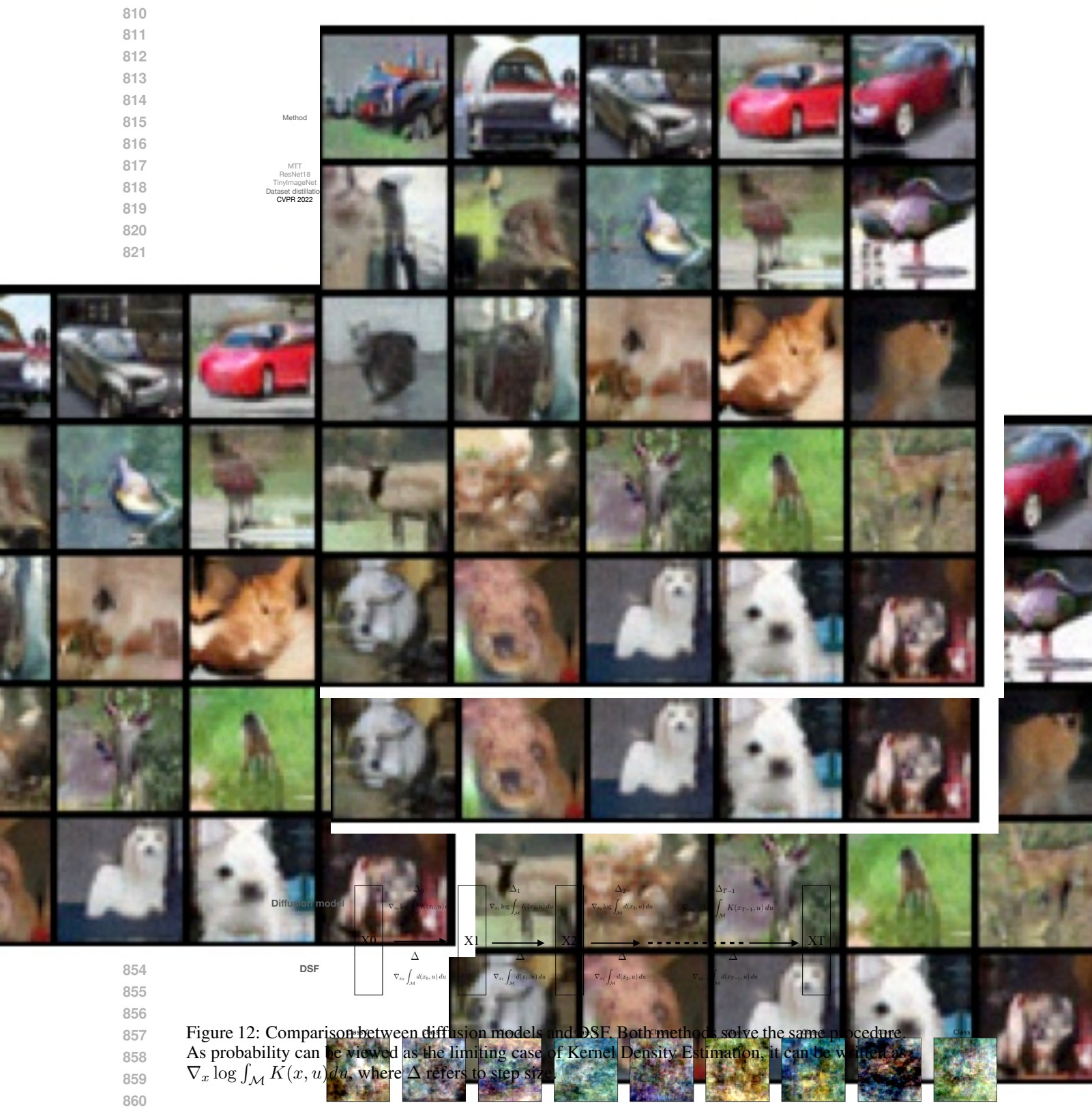

Figure 12: Comparison between diffusion models and DSF. Both methods solve the same procedure. As probability can be viewed as the limiting case of Kernel Density Estimation, it can be written as $\nabla_x \log \int_{\mathcal{M}} K(x, u) du$, where $\Delta$ refers to step size.

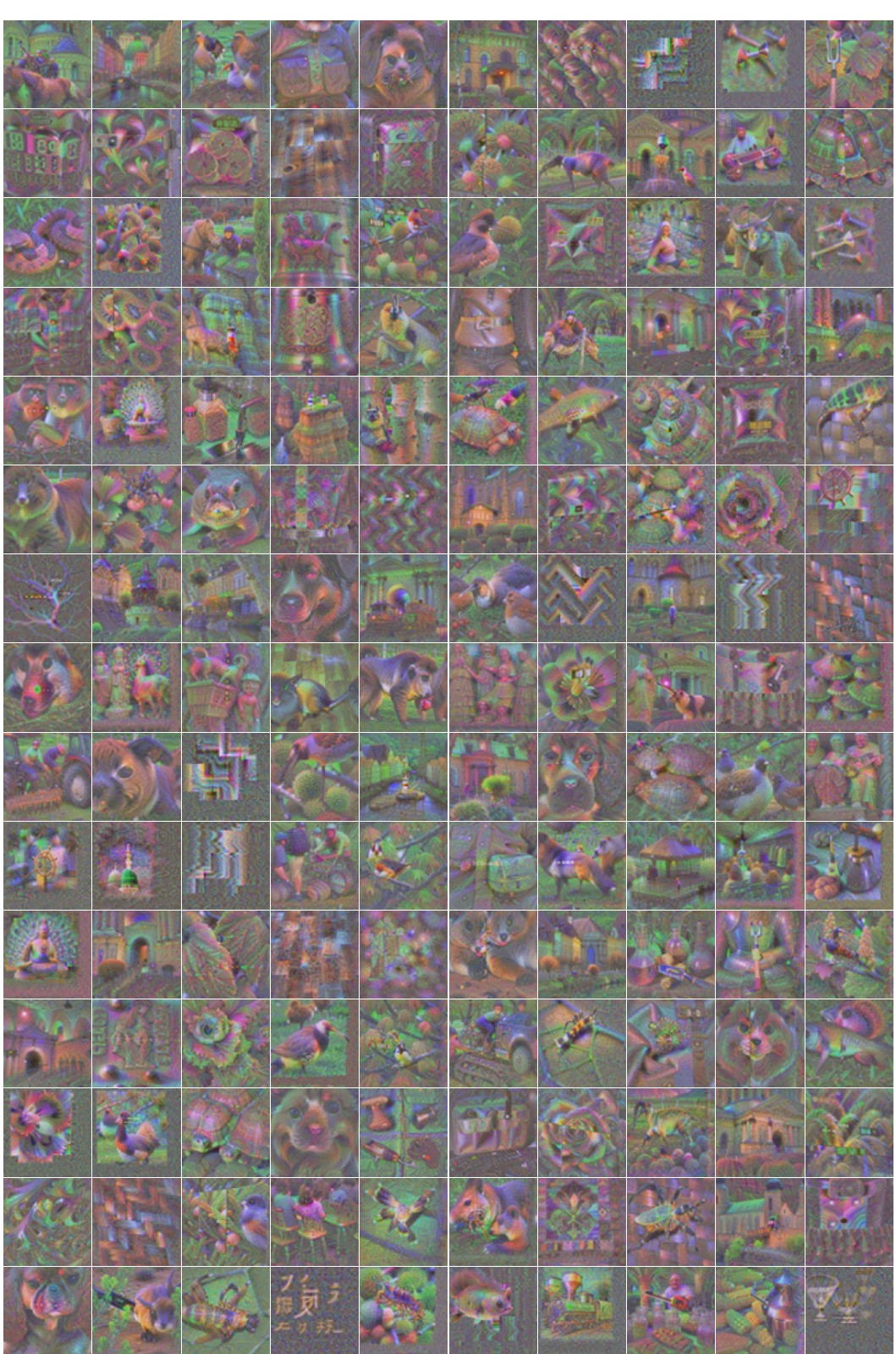

Figure 13: uncurated images of DSF (1)

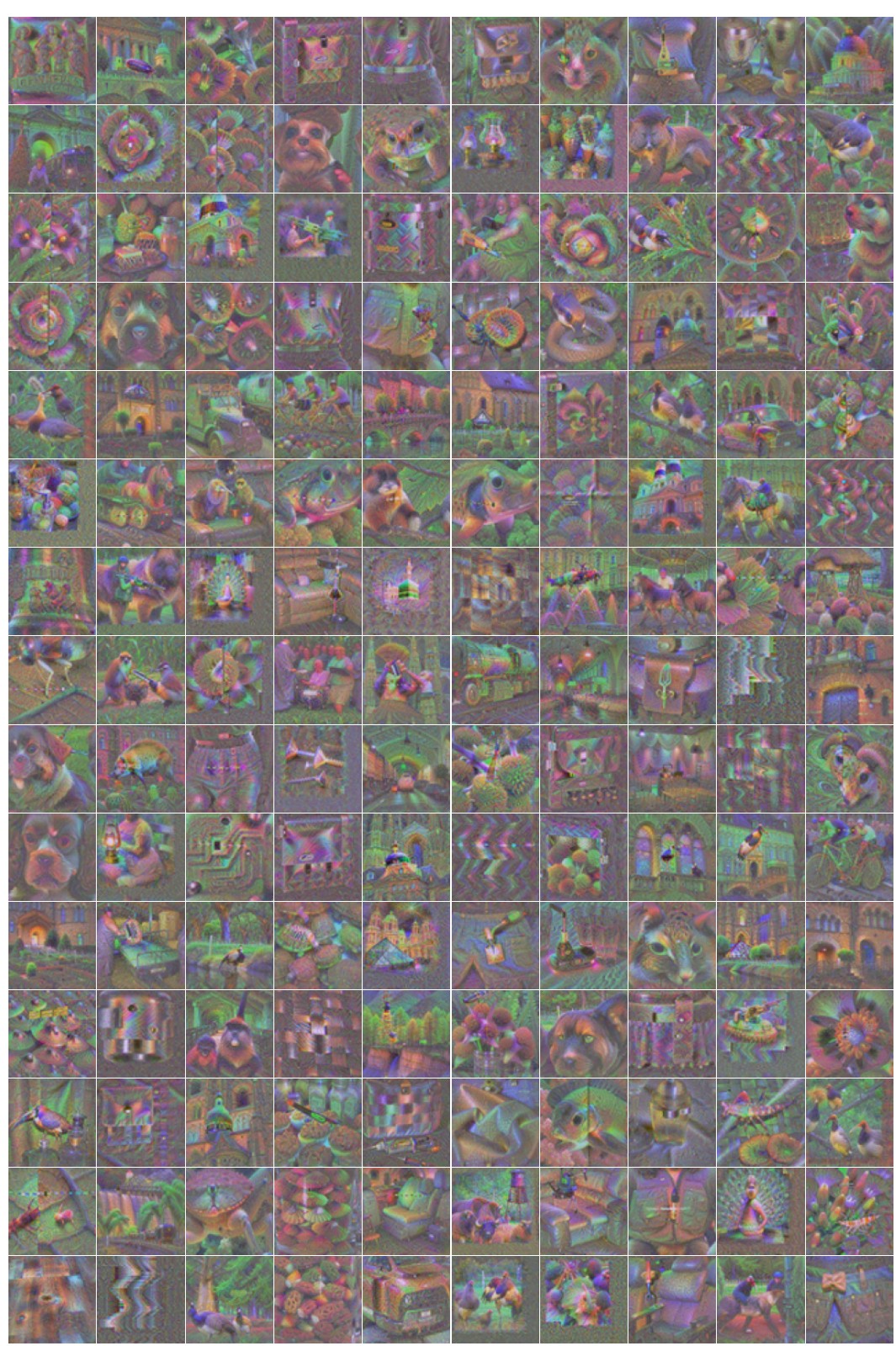

Figure 14: uncurated images of DSF (2)

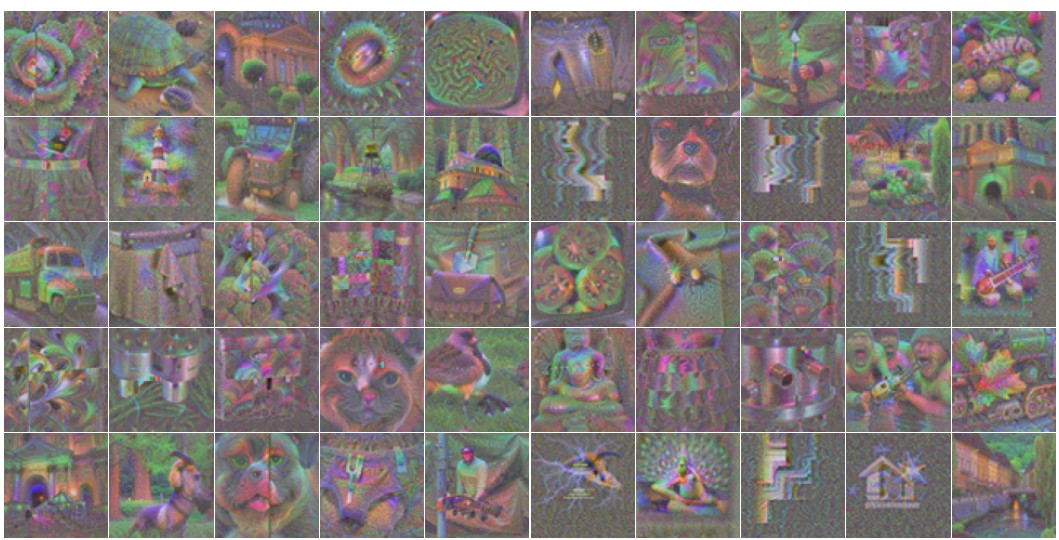

Figure 15: uncurated images of DSF (3)