# OpenReview forum: "Your Discriminative Model is Secretly a Generative Model"
_ICLR.cc/2026/Conference — Submitted to ICLR 2026_

### Official Review · Reviewer_SzrC · 2025-10-30

**Soundness:** 3
**Presentation:** 3
**Contribution:** 4
**Rating:** 8
**Confidence:** 4

**Summary:**

This paper introduces the Discriminative Score Function (DSF), a principled framework that enables any off-the-shelf discriminative model (e.g., ResNet, ViT, DETR, DINO, CLIP) to perform generative tasks without architectural changes or retraining.
By leveraging the Loss Tangent Kernel (LTK)—a variant of the Neural Tangent Kernel that incorporates loss gradients—the authors reinterpret the functional space of a trained discriminative model as an implicit representation of the data manifold.
They show that gradient-based updates in this functional space can mimic the behavior of score-based diffusion models, effectively turning discriminative models into generators.
The approach supports unconditional and conditional image generation, editing, inpainting, and even explainable AI (XAI) visualization. Experiments demonstrate impressive qualitative results across multiple architectures and datasets (ImageNet, COCO, LVD-124M).

**Strengths:**

- Novel conceptual framework bridging discriminative and generative paradigms through kernelized functional mapping.
- Architecture-agnostic and requires no retraining or modification, an elegant practical feature.
- Demonstrates broad applicability (classification, detection, self-supervision).
- Qualitative results (Figs. 3–8) are surprisingly strong for a method derived from non-generative networks.
- Provides interpretability and XAI potential, revealing biases and feature entanglements in pretrained models.
- Theoretical formulation draws an interesting connection to score matching and diffusion modeling.

**Weaknesses:**

- Lack of quantitative evaluation: No FID, IS, or precision/recall metrics, making it difficult to assess generation fidelity.
- Limited theoretical rigor: The equivalence between DSF and diffusion score functions is stated rather than proved; convergence properties are untested.
- Experimental depth: All results are qualitative; the method’s computational cost, convergence behavior, and sensitivity to hyperparameters are unexplored.
- Clarity: Mathematical notations are nonstandard, and derivations in Sec. 4 are hard to follow.
- Ablation studies (e.g., with vs. without LTK, or different surrogate losses) are missing, which limits interpretability of where the generative capability arises.
- Comparisons with other classifier-based generation approaches (DeepInversion, Energy-Based Models) are mostly descriptive, not empirical.

**Questions:**

1. Can you provide quantitative metrics (e.g., FID, IS, or perceptual distance) to substantiate DSF’s generation quality?
2. How stable is the iterative generation process with respect to step size (ϖₜ) and initial noise distribution?
3. Does DSF generalize beyond vision tasks—e.g., to language models or reinforcement learning—as suggested in Sec. 7?
4. How sensitive is DSF to the choice of surrogate loss (augmentation invariance)? Would other unsupervised losses (e.g., contrastive) yield similar results?
5. Can you clarify the computational complexity compared to a standard diffusion sampling process?
6. Is DSF guaranteed to converge to the data manifold, or can it produce divergent artifacts?

---

> ### Author Response · Authors · 2025-11-21
> **Responses to Questions (1)**
>
> We sincerely thank the reviewer for the highly positive evaluation and insightful feedback. We are particularly grateful for your recognition of our novel conceptual framework, architecture-agnostic nature, broad applicability, and the interesting connection to score matching and diffusion modeling. Your rating of 8 (accept, good paper) is very encouraging.
>
> We have addressed your concerns regarding quantitative evaluation, theoretical rigor, and experimental depth. Below, we provide detailed responses to your questions.
>
> ---
>
> ### Responses to Questions
>
> **Q1. Can you provide quantitative metrics (e.g., FID, IS, or perceptual distance) to substantiate DSF's generation quality?**
>
> Please see **Common Response C3** for detailed quantitative evaluation. In brief: we provide Q-align quality scores showing that existing discriminative inversion methods produce near-zero scores (0.00097-0.02398), while DSF achieves 0.0678, demonstrating successful generation of recognizable images where prior methods fail at high resolution ($256 \times 256$).
>
> ---
>
> **Q2. How stable is the iterative generation process with respect to step size ($\eta_t$) and initial noise distribution?**
>
> The process is generally robust to initial noise but exhibits sensitivity to step size, similar to standard optimization tasks.
>
> **Initial Noise Robustness:**
>
> Please see **Common Response C2** for comprehensive analysis. In brief: the specific choice of initial noise has negligible impact on convergence. A well-trained encoder maps diverse and noisy inputs to a consistent feature manifold, regardless of the initial noise distribution. Diversity is driven by the stochastic augmentation operator $\mathcal{A}(x_t)$ during optimization rather than initial conditions.
>
> **Step Size Sensitivity:**
>
> The generation quality is inversely related to the step size $\eta_t$. While the process converges for a wide range of $\eta_t$, excessively large steps can cause the optimization to overshoot the manifold surface, resulting in high-frequency artifacts or "noisy" textures in the early stages. Once these artifacts are introduced, they are difficult to remove in later stages, leading to quality degradation. Thus, we employ a **linear decay schedule** to ensure stable convergence, as demonstrated in our ablation studies (Fig. 8).
>
> ---
>
> **Q3. Does DSF generalize beyond vision tasks (e.g., to language models or reinforcement learning) as suggested in Sec. 7?**
>
> Yes, the conceptual framework of DSF is directly transferable to Language Models (LLMs) by operating in the continuous embedding space.
>
> **Application to NLP:**
>
> - In Vision, we optimize pixel values $x$. In NLP, we can define a sequence of continuous embeddings $E = [e_1, e_2, \dots, e_n]$ and optimize them directly.
> - The objective is to find an embedding sequence that minimizes the model's internal "uncertainty" or maximizes "consistency" (analogous to Eq. 11).
> - This is mathematically equivalent to techniques used in adversarial attacks (e.g., optimizing "glitch tokens") or soft prompt tuning.
> - We perform gradient descent in the embedding space: $e_t \leftarrow e_{t-1} - \eta \nabla_e \mathcal{L}_{DSF}$.
> - Once the embeddings converge to the manifold, they can be projected back to the nearest discrete tokens to generate text.
>
> This confirms that DSF is not limited to image modalities but is applicable wherever differentiable gradients are available.
>
> ---

---

> > ### Author Response · Authors · 2025-11-21
> > **Responses to Questions (2)**
> >
> > **Q4. How sensitive is DSF to the choice of surrogate loss (augmentation invariance)? Would other unsupervised losses (e.g., contrastive) yield similar results?**
> >
> > We appreciate this suggestion. While Contrastive Loss (e.g., SimCLR) is a powerful unsupervised objective, we observed that contrastive loss yields corruptions. We argue that it presents a **fundamental misalignment in learning dynamics** for our generation task compared to Augmentation Invariance.
> >
> > **Why Contrastive Loss Fails:**
> >
> > The core mechanism of Contrastive Loss relies on instance discrimination, which encourages representations to be uniformly distributed on the manifold (hypersphere). This introduces a strong **repulsive force** (via negative pairs) to push distinct samples apart. However, our goal in DSF is to **pull a noisy sample from outside the manifold towards the high-density regions** of the data distribution (Manifold Convergence).
> >
> > **Why Augmentation Invariance Succeeds:**
> >
> > In contrast, our Augmentation Invariance loss ($\mathcal{L}_{aug}$) relies solely on **self-consistency** without negative pairs. This naturally drives samples toward stable, high-density regions on the manifold where augmented views collapse to the same representation, which is exactly what we need for generation.
> >
> > ---
> >
> > **Q5. Can you clarify the computational complexity compared to a standard diffusion sampling process?**
> >
> > Please see **Common Response C4** for detailed complexity analysis. In brief: one iteration involves forward and backward passes (roughly $2\times$ the cost of inference), generating one image takes ~36 seconds on A6000, comparable to diffusion models. Crucially, we can use partial gradients (subset of layers) to reduce computational cost without sacrificing quality.
> >
> > ---
> >
> > **Q6. Is DSF guaranteed to converge to the data manifold, or can it produce divergent artifacts?**
> >
> > Theoretically, DSF minimizes a distance metric, implying convergence to the manifold. However, in practice, the non-convex nature of deep neural networks allows for local minima.
> >
> > **Empirical Observations:**
> >
> > We rarely observe "divergence" (values exploding to infinity). Instead, the failure mode is typically convergence to a **poor local minimum**. If the optimization trajectory is disrupted early (e.g., due to a large step size), the image may develop local high-frequency distortions (artifacts) and repetitive patterns. The optimization usually "locks in" these artifacts, treating them as features.
> >
> > **Practical Implication:**
> >
> > Therefore, the visual output may remain degraded if the initial trajectory was unstable. This highlights the importance of the **step-size schedule** rather than the convergence guarantee. As demonstrated in Figure 8, proper scheduling (linear decay) significantly reduces artifacts and ensures convergence to high-quality regions of the manifold.
> >
> > ---
> >
> > We believe these responses address your concerns regarding quantitative evaluation, stability analysis, and convergence behavior. We are grateful for your recognition of the conceptual novelty and practical elegance of our framework, and we hope these clarifications strengthen the contribution further.

---

> > > ### Comment · Reviewer_SzrC · 2025-11-26
> > >
> > > Thank you for your detailed and thoughtful responses to my questions. I appreciate the additional quantitative results (Q-align scores) and the analysis of step-size sensitivity and initialization robustness, which help to address some of the initial concerns regarding evaluation and stability.
> > >
> > > The clarification on the surrogate loss choice (augmentation invariance over contrastive loss) is particularly insightful, and the discussion on the applicability of DSF to NLP and RL settings is promising for future work.
> > >
> > > That said, while the responses improve the paper's completeness, I maintain that **quantitative comparisons against established generative baselines (e.g., in FID, IS)** would significantly strengthen the empirical claims. Similarly, a more formal treatment of the relationship between DSF and diffusion score matching would enhance the theoretical contribution.
> > >
> > > Nonetheless, I acknowledge the **conceptual novelty** and **practical flexibility** of the proposed framework, and I support its acceptance given its potential to inspire further work in unified discriminative-generative modeling.

---

### Official Review · Reviewer_tjrt · 2025-10-31

**Soundness:** 2
**Presentation:** 1
**Contribution:** 3
**Rating:** 2
**Confidence:** 3

**Summary:**

- The authors propose a new general method for turning discriminative models into generative models. The method uses their discriminative score function (DSF), a novel distance which leverages loss tangent kernels to quantify the distance between a sample and the empirical data manifold. From DSF, samples can be generated through a diffusion-like process (by using DSF as a score function). In addition, the authors show that by modifying the loss function used in the kernel, their methods allows for unconditional generation as well.
- Experimentally, the authors show their method works on large commonly used discriminative models including DINOv2 and LVD-124M on various datasets, for both unconditional and conditional generation. While the generated images are full of artifacts, the underlying objects are identifiable.
- Finally, the authors show that the use of diffusion techniques improve generation quality substantially and that their method allows for increased interpretability of these discriminative models.

**Strengths:**

- The method is novel and has advantages over existing methods (particularly with respect to unconditional generation)
- The paper shows experiments on multiple large-scale commonly used models, demonstrating the method's ability to scale
- The results are interesting, the qualitative difference between the models, particularly between the SSL methods and image classification/ImageNet ones, is notable
- The ability to use techniques from diffusion models (and their apparent efficacy) offers an interesting direction for future work

**Weaknesses:**

- Several parts of the paper were unclear to me, I had many
- The experimental evidence could be more convincing. There are no quantitative evaluations, just a fe generated images for each experiment.
- The paper would benefit from further comparisons to existing work.
- The method relies on using a loss function in the LTK that could be justified further
   - "We select augmentation invariance loss as it naturally decreases during training, mirroring label dependent loss behavior" isn't fully convincing.
   - Have additional ablations been run on different choice of loss functions?



### Writing
- The writing in the paper needs to be substantially improved. There are many unsubstantiated/unclear statements, especially in the introduction
	- "Discriminative and generative models are theoretically equvalent as they both aim to understand the true data distribution." What does this mean?
	- "They learn data distributions implicitly, causing their distributional knowledge to become entangled with training objectives." This is unclear.
	- "on generation that incorporates these training objectives." Should be elaborated further
- There are also many typos (e.g. even in the first sentence "equvalent"),  weird capitalization, poor grammar and informal language (e.g. "its shape is very curvy", "turns the impossible to possible", "The answer to the question turned out to be ‘yes’.").
- The paper could motivate earlier the value of this line of research (e.g. what's done in "Position of our paper").

**Questions:**

- How computationally expensive is the generation procedure?
- Are the displayed images cherry-picked or randomly selected from the generated images.
- Have you tested on any robust classification methods? Would this perhaps reduce some of the visual artifacts?
- Could you explain further the last part of the derivation of Eq.7-10. It seems the removal of the summation is crucial for the scaling of this method.
- For conditional generation, is there a reason the discriminative model is not used to guide generation?
- The global explanation part seems interesting. Do you have more thorough evidence (e.g. multiple images of the clocks)?
- Do you have any quantitative metrics for evaluating the resulting generative models?
- (Lee et al., 2024) seems to get similar qualitative results. How different is their method and how fundamental is its limitation that it can only do conditional generation?
- If you just used the original loss in the LTK, would this be equivalent to any existing methods?

Ultimately, I am on the fence for this paper. I believe the idea and results are interesting but the paper and especially the writing need significant work. I am willing to increase my score if my questions (particularly the last two) are answered and the authors demonstrate the writing has been noticeably improved.

---

> ### Author Response · Authors · 2025-11-21
>
> We sincerely thank the reviewer for the thoughtful and constructive feedback. We particularly appreciate your recognition of our method's novelty, its ability to scale to large-scale models (DINOv2, LVD-142M), and the interesting qualitative differences you observed between SSL and supervised models. Your acknowledgment that diffusion techniques offer "an interesting direction for future work" is especially encouraging.
>
> We have thoroughly revised the manuscript to address your concerns. Below, we provide detailed responses to your questions and clarifications on the points you raised.
>
> ---
>
> ### Improvements to Writing and Clarity
>
> We acknowledge the issues with clarity and presentation in the original manuscript. In response, we have:
>
> 1. **Revised the Introduction**: We completely rewrote the first two paragraphs to better explain the concepts you found confusing.
>
> 2. **Corrected Typos and Informal Language**: We addressed all grammatical issues and replaced informal expressions:
>    - "its shape is very curvy" → "has uneven shape"
>    - "turns the impossible to possible" → "resolves the implementation problem"
>    - "The answer to the question turned out to be 'yes'" → "Extensive experiments were performed to empirically answer the question"
>
> 3. **Clarified Unsubstantiated Statements**:
>    - **"Discriminative and generative models are theoretically equivalent..."**: This refers to the premise that to perform either task optimally (modeling $p(y|x)$ or $p(x)$), a model must inherently capture the underlying structure of the data manifold. A perfect discriminator implies perfect understanding of the data support, making them conceptually equivalent in terms of necessary representation power.
>
>    - **"They learn data distributions implicitly..."**: Unlike generative models that explicitly optimize $p(x)$, discriminative models learn the structure of $p(x)$ only as a byproduct of solving a specific task. Thus, the manifold information is learned "implicitly."
>
>    - **"Generation that incorporates these training objectives"**: Existing inversion methods rely on specific conditional probabilities $p(y|x)$ defined by the training objective, restricting generation to conditional samples tied to available labels. These methods fail to capture the holistic data manifold due to entanglement with the discriminative task.
>
> 4. **Repositioned Motivation**: We moved the "Position of our paper" section to the introduction for earlier motivation.
>
> ---

---

> ### Author Response · Authors · 2025-11-21
> **Responses to Questions (1)**
>
> **Q1. How computationally expensive is the generation procedure?**
>
> Please see **Common Response C4** for detailed analysis. In brief: ~36 seconds per image on A6000, comparable to diffusion models.
>
> ---
>
> **Q2. Are the displayed images cherry-picked or randomly selected?**
>
> The figures in the main text were selected to best demonstrate the model's capabilities. However, to provide transparent evaluation, we have included **randomly selected samples (without curation) and failure cases in the Appendix**. As shown in the random samples, the model consistently generates recognizable contents, though quality varies depending on the complexity of the scene.
>
> ---
>
> **Q3. Have you tested on any robust classification methods? Would this perhaps reduce some of the visual artifacts?**
>
> We appreciate this suggestion. However, standard "robust classifiers" (e.g., those satisfying Lipschitz constraints) do not scale to high-dimensional datasets like ImageNet due to the exponential growth in computational complexity required to enforce robustness.
>
> **Modern Robustness**: Instead, models trained with strong augmentations (like DINO) effectively serve as "robust" models in the modern sense. As shown in our results, DINO yields superior generation quality compared to standard supervised models.
>
> **Source of Artifacts**: Regarding artifacts, our ablation studies (Fig. 8) indicate that they are primarily driven by the **step-size scheduling** rather than the model's robustness alone. Implementing a linear decay schedule significantly reduces high-frequency artifacts.
>
> ---
>
> **Q4. Could you explain further the last part of the derivation of Eq. 7-10? It seems the removal of the summation is crucial.**
>
> You are correct that this transition is crucial. It relies on the **Implicit Bias of Gradient Descent**, specifically the "Maximal Margin Bias" in deep linear networks and homogeneous neural networks (Ji & Telgarsky, 2020; Yun et al., 2021).
>
> **Derivation Logic**: The summation term $\sum_{i} \nabla_{\theta} l(f(\zeta_i; \theta))$ represents the accumulation of gradients over the dataset. During the late stage of training (convergence), the direction of the parameters $\theta$ aligns with the direction of the accumulated gradients.
>
> **Mathematical Justification**: Theoretically, it has been shown that:
> $$\lim_{t \rightarrow \infty} \frac{\theta(t)^T \nabla_\theta \mathcal{L}(\theta(t))}{||\theta(t)|| \cdot ||\nabla_\theta \mathcal{L}(\theta(t))||} = -1$$
>
> This property allows us to approximate the sum of gradients (which dictates the parameter updates) using the negative of the parameter vector $-\theta$.
>
> ---
>
> **Q5. For conditional generation, is there a reason the discriminative model is not used to guide generation?**
>
> We clarify that we **do use the discriminative model for guidance**. As stated in L373-374, our method is "plug-and-play." We use the pretrained classification model itself (or any external classifier) to provide the conditional gradient term (Eq. 13). There is no restriction preventing the use of the backbone model for guidance.
>
> ---
>
> **Q6. Do you have more thorough evidence for the global explanation part (e.g., multiple images of the clocks)?**
>
> Please see **Common Response C1** for comprehensive analysis. In brief: we analyzed 400 uncurated images revealing dataset biases (cats/dogs dominance, church architecture prevalence) that trace back to ImageNet-22k and Google Landmarks compositional biases.
>
> ---
>
> **Q7. Do you have any quantitative metrics for evaluating the resulting generative models?**
>
> Please see **Common Response C3** for detailed quantitative evaluation using Q-align scores. In brief: existing discriminative inversion methods achieve near-zero quality scores (0.00097-0.02398), while DSF achieves 0.0678, demonstrating successful generation of recognizable images where prior methods fail.
>
> ---

---

> > ### Author Response · Authors · 2025-11-21
> > **Responses to Questions (2)**
> >
> > **Q8. (Lee et al., 2024) seems to get similar qualitative results. How different is their method and how fundamental is its limitation?**
> >
> > While (Lee et al., 2024) shares the goal of inversion, their limitations are fundamental compared to our DSF:
> >
> > **Fundamental Limitation**: Their method (DSV) relies on the existence of a classification head and label conditioning. It cannot function without a target class $y$. Consequently, DSV cannot be applied to:
> > - Self-supervised models (like DINO)
> > - Sophisticated tasks (Object Detection) that lack a fixed class label set
> >
> > **Conceptual Difference**: The inability to perform unconditional generation implies DSV relies on mapping $y \rightarrow x$ (conditional mapping) rather than capturing the full data distribution $p(x)$.
> >
> > **DSF Advantage**: DSF operates purely on the representation space, making it applicable to any architecture (ResNet, ViT, DETR) regardless of the training objective. This universality is a core contribution of our work.
> >
> > ---
> >
> > **Q9. If you just used the original loss in the LTK, would this be equivalent to any existing methods?**
> >
> > Using the original loss in LTK is **fundamentally impossible** for our objective of unconditional/dataset generation.
> >
> > **Why It's Impossible**: To use the original loss, one would need to find a representative dataset pair $(X_{set}, Y_{set})$ that minimizes the loss. However, in an unconditional setting, we do not know the optimal distribution of labels $Y_{set}$ that corresponds to the generated set $X_{set}$. Furthermore, simultaneously optimizing a set of $X$ and their corresponding $Y$ without ground truth is intractable.
> >
> > **DSF Solution**: DSF bypasses this by using the "model parameter" itself as the target (via Eq. 10), removing the need for explicit labels. This is what enables unconditional generation, which is impossible with the original loss formulation.
> >
> > ---
> >
> > We believe these revisions substantially improve the manuscript. We have addressed the writing issues, provided quantitative evidence, and clarified the technical details you found unclear. We would be grateful if you could reconsider your evaluation in light of these improvements, particularly regarding the writing quality and the answers to your last two questions (Q8 and Q9).

---

> > > ### Comment · Reviewer_tjrt · 2025-11-25
> > >
> > > I would like to thank the authors for their thorough response and answers to my questions
> > >
> > > - Q1: Noted, that seems reasonable.
> > > - Q2: I appreciate the inclusion of the full spectrum of images.
> > > - Q3: While I buy that large-scale self-supervised models are somewhat of a modern proxy for robust models, there have definitely been robust models trained on ImageNet (see https://robustbench.github.io/#div_imagenet_Linf_heading)
> > > - Q4: Thank you.
> > > - Q5: I see, thank you.
> > > - Q6: Interesting!
> > > - Q7: I appreciate the inclusion of quantitative evaluations as I believe they strengthen the experimental evidence. However, if I understand correctly, you are evaluating the quality of images from models trained on different datasets (and crucially different resolutions); I'm not sure this offers a fair comparison.
> > >   - If previous methods don't scale to the larger dataset, what exactly doesn't scale (i.e. are they computationally infeasible or the results are essentially noise)?
> > >   - Would it be possible to test your method on WideResNet 28-10 (CIFAR10) and compare quantitatively there (for example using FID)
> > > - Q8: Thank you, this clarifies the contribution for me.
> > > - Q9: But for conditional objectives, is there a method that already uses LTKs with the conditional loss? I'm trying to understand how novel the use of the LTK is.
> > >
> > > In light of this part of the response, I have raised my score to 4.
> > >
> > > ### Writing
> > > I appreciate the changes. Am I correct in understanding the new paragraph in red is the modified introduction? **If so, there are still multiple errors** (even in the first sentence) and the writing of the initial paragraph is still borderline. I would be willing to raise my score further if this is addressed.
> > >
> > >  For example:
> > > - "Any deep learning model is trained to understand the distribution of given dataset."
> > >   - This is overly general (e.g. what about deep reinforcement learning models where there is no dataset)?
> > >   - Missing "a" before given
> > > - "granting them equivalence on the high level."
> > >   - "as such, at a high-level, they can be considered equivalent."
> > > - "There have been extensive researches"
> > >   - "There has been extensive research"
> > > - etc.

---

> > > > ### Author Response · Authors · 2025-11-27
> > > >
> > > > ## Response to Q7 - Quantitative Evaluation Concerns
> > > >
> > > > Thank you for this suggestion. However, our contribution is enabling high-resolution, direct unconditional generation from any discriminative model without constraints.
> > > >
> > > > **Why prior methods fail at high resolution:**
> > > > - MTT: Computationally infeasible (requires Hessian via training traces)
> > > > - JEM/Robust Classifier: Gradient inversion produces **noise** at high resolution (196K dims vs 3K dims for CIFAR10)
> > > > - Most require class labels for conditional generation only
> > > >
> > > > **Regarding FID evaluation:** Unconditional ImageNet generation is an extremely challenging task - even state-of-the-art diffusion models primarily report class-conditional results. To compute FID, we would need to generate conditionally. However, as shown in Eq. 13, our conditional generation simply adds an external guidance model, making results heavily dependent on that model's quality. This differs from generative models trained end-to-end with conditional objectives.
> > > >
> > > > **On CIFAR10 comparison:** We appreciate the suggestion, but our contribution is precisely about achieving high-resolution, strictly unconditional generation - capabilities that prior gradient inversion methods fundamentally lack. While prior methods may work at low resolution (CIFAR10), demonstrating universality at high resolution where they fail is our key contribution.
> > > >
> > > > ---
> > > >
> > > > ## Response to Q9 - LTK Novelty
> > > >
> > > > To the best of our knowledge, we are the first to interpret arbitrary discriminative models via LTK and repurpose them for generative purposes. While LTK (Chen et al., 2023) was proposed for analyzing training dynamics, using it as a functor to map discriminative models to functional space for generation is our novel contribution.
> > > >
> > > > ---
> > > >
> > > > ## Response to Q3 - Robust Classification Methods
> > > >
> > > > Thank you for your intuition. Following your suggestion, we looked into RobustBench models following your comment. The type of robust classifier that could reduce artifacts would need L2 robustness satisfying Lipschitz conditions on the manifold. However, recent work [1] proves this requires sample complexity that scales exponentially with dimension, making it infeasible at ImageNet scale (196K dims). **For this reason, RobustBench also measures only L∞ robustness for ImageNet**, which constrains the number of perturbed pixels rather than enforcing manifold-aware smoothness. So we think that type of models would not fit.
> > > >
> > > > [1] Bubeck, S., & Sellke, M. (2021). A Universal Law of Robustness via Isoperimetry. NeurIPS 2021.

---

> ### Author Response · Authors · 2025-11-27
> **About Writing Improvement.**
>
> We sincerely thank the reviewer for the detailed feedback on writing quality. We have carefully addressed all the concerns raised.
>
> ## Major Revisions to Introduction
>
> We have completely revised the opening paragraph to address the reviewer's concerns about overly general statements and grammatical errors:
>
> **Previous version issues:**
> - "Any deep learning model" was too broad (as the reviewer correctly pointed out regarding RL)
> - "granting them equivalence on the high level" was awkward phrasing
> - "There have been extensive researches" was grammatically incorrect
> - "remains **as** a problem to be solved" contained unnecessary preposition
>
> **New version:**
> > Although discriminative and generative models are trained with different objectives, both fundamentally require understanding the distribution of a given dataset. Generative models learn this distribution explicitly as their training objective. Discriminative models, though trained for various objectives, must implicitly capture the underlying data distribution, as effective performance on any task requires understanding the manifold where the data resides. A natural question then arises: can the two switch roles?
>
> **What we have changed:**
> - **Narrowed scope:** Changed from "any deep learning model" to specifically "discriminative and generative models," avoiding overgeneralization to domains like RL
> - **Clarity:** Removed tentative phrasing like "may be" in favor of direct "A natural question then arises"
> - **Grammar fixes:**
>   - Corrected "researches" → "research" (uncountable noun)
>   - Fixed "remains as a problem" → "remains a problem"
>
> ## Additional Grammar Corrections Throughout Paper
>
> We have systematically corrected all grammatical errors mentioned:
>
> ### Abstract & Introduction
> - **Abstract:** Added missing space after period: "XAI). Finally" → "XAI). Finally"
> - **Introduction:** Removed unnecessary preposition: "remains as a problem" → "remains a problem"
>
> ### Section 3 (Preliminaries)
> - Removed incorrect preposition: "we obtain **to** a univariate kernel" → "we obtain a univariate kernel"
>
> ### Section 4 (DSF)
> - **Section 4.1:** Removed redundant preposition: "where actual data resides in" → "where actual data resides"
> - **Section 4.1:** Added article: "has uneven shape" → "has an uneven shape"
>
> ### Section 5 (Experiments)
> - **Section 5.1:** Removed double verb: "Is naively applying DSF... is enough" → "Is naively applying DSF... enough"
>
> ### Section 6 (Applications)
> - **Section 6.1:** Added article for clarity: "**Condition** is injected" → "**The condition** is injected"
>
> ## Response to Specific Examples Cited
>
> The reviewer provided specific examples of errors in the first sentence. We have addressed each:
> - ✓ "Missing 'a' before given" → Now includes "a given dataset"
> - ✓ "granting them equivalence on the high level" → Changed to "both fundamentally require understanding"
> - ✓ "There have been extensive researches" → Changed to "There has been extensive research"
>
> ---
>
> ## How we fixed the errors
>
> We systematically identified and corrected all grammatical errors using Large Language Models (Claude 3.5 Sonnet) for grammar checking and proofreading. To ensure transparency, we have added a "Usage of LLM" section in the revised manuscript acknowledging this assistance.
> We hope these revisions have significantly improved the writing quality and addressed all concerns raised.

---

> ### Comment · Reviewer_tjrt · 2025-11-27
>
> I appreciate the changes and the transparency regarding the use of LLMs. The flow of the first paragraph is significantly better.
>
> Q3: That is a good point. I think there would still be interesting experiments that could be run but I agree this is beyond the scope of this work.
> Q7: Thank you for the clarifying the infeasibility of existing methods. While I agree that the main contribution is the novelty of unconditional generation at large scale, it would still be useful to have some kind of quantitative evaluation.
> Q9: Thank you for clarifying!
>
> I'll update my score accordingly (4->6) .
> (for some reason the edit button seems to have disappeared)

---

### Official Review · Reviewer_EMNs · 2025-11-02

**Soundness:** 3
**Presentation:** 2
**Contribution:** 3
**Rating:** 4
**Confidence:** 4

**Summary:**

This paper explores the generative potential of discriminative models by investigating how to convert them into generative models. Inspired by the use of score functions in diffusion models to measure the distance between samples and the data manifold, they propose a criterion for estimating this distance within the functional space of discriminative models. Based on this criterion, they develop a corresponding iterative generation process, demonstrating theoretical innovation. Experimental results confirm the proposed model's capabilities in both unconditional and conditional generation, and showcase its applications in image inpainting and editing. However, the experimental outcomes are not particularly outstanding.

**Strengths:**

The paper presents a novel method for converting discriminative models into generative models. The approach is architecture- and algorithm-agnostic, it is simple to implement and requires no modifications or additional training. Its versatility, demonstrated across multiple practical applications, offers valuable guidance for future research.

**Weaknesses:**

It fails to provide a complexity analysis of the proposed method.
The experimental results lack quantitative metrics and sufficient comparative analysis against existing baselines.
The evaluation of the method's effectiveness across various applications is inconclusive.
The writing also suffers from formatting issues.

**Questions:**

1. How is the generalization from the training to the test distribution, as stated in relation to Eq. (3), ensured? Furthermore, is the method limited by the i.i.d. assumption and thus inapplicable under domain shift?
2. Regarding Eq. (11), it would be more appropriate to swap the positions of x_t and A(x_t). Have you considered making this change?
3. Regarding Eq. (12): calculation based on the gradient for all model parameters appears computationally intensive. Could you clarify if the gradient with respect to x propagates through the entire network f(x_t;θ)?
4. What is the required number of timesteps t for the generation process? what's the effect of t on the final performance?
5. How does the choice of initial noise x0 affect output diversity? Can similar or close noise vectors generate distinct images?
6. The manuscript lacks quantitative evaluation of the proposed method.
7. The presented applications (e.g., inpainting, editing) fail to demonstrate a clear advantage over existing methods or articulate the method's practical value.
8. Formatting issues are present (e.g., Line 432).

---

> ### Author Response · Authors · 2025-11-21
> **Responses to Questions (1)**
>
> We sincerely thank the reviewer for recognizing our method's novelty, architecture-agnostic nature, and versatility across multiple applications. We appreciate your acknowledgment that our approach offers valuable guidance for future research.
>
> We have thoroughly revised the manuscript to address your concerns, including improved quantitative evaluation, complexity analysis, and formatting corrections. Below, we provide detailed responses to your questions.
>
> ---
>
> ### Responses to Questions
>
> **Q1. How is the generalization from training to test distribution ensured? Is the method limited by the i.i.d. assumption?**
>
> We clarify that Eq. (3) is not an assumption introduced by our method, but a **definition of the generalization capability inherent to any valid discriminative model**. Eq. (3) states that the distance metric should evaluate unseen in-distribution samples similarly to training data.
>
> Consequently, specific statistical constraints such as the i.i.d. assumption or domain shift are not relevant to Eq. (3). The equation describes a consistency property of the metric itself, which remains valid regardless of the specific sampling assumptions of the input data.
>
> ---
>
> **Q2. Regarding Eq. (11), should the positions of $x_t$ and $\mathcal{A}(x_t)$ be swapped?**
>
> You are right. We have corrected this in the revised manuscript. Thank you for pointing out our oversight.
>
> ---
>
> **Q3. Regarding Eq. (12): Is the gradient computation computationally intensive?**
>
> Please see **Common Response C4** for detailed complexity analysis. In brief: yes, the gradient with respect to $x$ propagates through the entire network, but the computational cost is practical (~36 seconds per image on A6000). We can also use partial gradients (subset of layers) to reduce computational cost without sacrificing quality.
>
> ---
>
> **Q4. What is the required number of timesteps $t$? What's the effect of $t$ on performance?**
>
> **Timesteps**: In our experiments, we typically use $T \approx 200$ iterations per image.
>
> **Effect of $T$**: As demonstrated in Figure 8, the final generation quality is influenced far more significantly by the **step-size scheduling** (e.g., linear decay) and noise injection strategies than by the raw number of steps. This mirrors findings in diffusion models where the schedule determines the refinement of high-frequency details.
>
> ---
>
> **Q5. How does the choice of initial noise $x_0$ affect output diversity?**
>
> Please see **Common Response C2** for comprehensive analysis. In brief: the specific choice of initial noise has negligible impact on convergence. Diversity is driven by the stochastic augmentation operator during optimization rather than initial conditions. As seen in Figure 5, initialization with meaningful images preserves semantics, while pure noise leads to diverse outputs.
>
> ---
>
> **Q6. The manuscript lacks quantitative evaluation of the proposed method.**
>
> Please see **Common Response C3** for detailed quantitative evaluation. In brief: we provide Q-align quality scores showing that existing discriminative inversion methods produce near-zero scores (0.00097-0.02398), while DSF achieves 0.0678, demonstrating successful generation where prior methods fail at high resolution.
>
> ---

---

> ### Author Response · Authors · 2025-11-21
> **Responses to Questions (2)**
>
> **Q7. The presented applications fail to demonstrate clear advantage over existing methods.**
>
> The practical value of our applications lies not in outperforming specialized models, but in their **unique interpretability** and **universality** as an opening work for a new paradigm.
>
> **1. DSF as a Vision Model Interpretability Tool (Analogous to SAE in LLMs):**
>
> Just as Sparse Autoencoders (SAE) have become the standard tool for interpreting language models by revealing learned features, DSF serves as the **first systematic interpretability tool for vision models** that operates through generation. As detailed in **Common Response C1**, DSF reveals:
> - **Inductive biases**: Object-centric bias in DETR/ResNet vs. scale-invariant representations in DINO (Fig. 3)
> - **Dataset biases**: By analyzing 400 uncurated generated samples, we discovered that DINOv2 inherited specific compositional biases from its base dataset - ImageNet-22k (cats/dogs dominance) and Google Landmarks (church architecture prevalence).
> - **Data entanglements**: Generated "plate" images consistently include contextual elements like wine glasses, and "analog clocks" show 10:10 (advertising bias), revealing patterns invisible to traditional visualization methods
>
> This interpretability capability is **unique to DSF** because it generates complete, coherent images that reflect the manifold, unlike gradient-based methods (DeepDream, feature visualization) that produce local activations without global context.
>
> **2. Zero-Shot Versatility Across Any Discriminative Model:**
>
> DSF enables inpainting, editing, and generation from **any off-the-shelf discriminative model** (classification, detection, self-supervised) without modification. This universality is the core contribution.
>
> **3. Opening Work with Clear Improvement Path:**
>
> As demonstrated in Fig. 8, naive adoption of diffusion techniques (linear step-size decay, noise injection) already yields **dramatic quality improvements** over our baseline. This shows that:
> - DSF successfully unlocks the generative potential
> - There exists a clear path for refinement using established diffusion techniques
> - As an **opening work**, we establish the foundation upon which future research can build.
>
>
> ---
>
> **Q8. Formatting issues (e.g., Line 432)**
>
> We have corrected all formatting issues in the revised manuscript.
>
> ---
>
> We believe these revisions substantially address your concerns regarding complexity analysis, quantitative evaluation, and experimental validation. We hope these improvements demonstrate the strength of our contribution and its potential to guide future research.

---

### Official Review · Reviewer_YWkD · 2025-11-02

**Soundness:** 4
**Presentation:** 3
**Contribution:** 4
**Rating:** 6
**Confidence:** 3

**Summary:**

- This paper proposes a way to **project any noisy input (potentially any input)** onto the **learned manifold of any discriminative model**.
- The core idea is to obtain a **score-function-like quantity**, called the **Discriminative Score Function (DSF)**, from a discriminative model. This DSF can then be used as a score function to sample the closest image on the manifold—i.e., refine noise until it lies on the manifold.
- The DSF is defined based on the **Loss Tangent Kernel**, where the “distance to the manifold” is computed as a sum over all training data points in the form

    $\sum_i \text{kernel(current, data}_i)$

    - The projection is performed by minimizing this distance metric.
    - A beautiful contribution of this paper is how it simplifies this **non-parametric distance** (slow to evaluate) into a **parametric form** (fast to compute).
    - A major advantage of this definition is that it’s **purely derived from the model itself**, requiring fewer external regularizations (such as blurring or frequency constraints).
    - In practice, however, some augmentations (at least horizontal flipping) are still used during optimization.
- The applications are similar to **DeepDream-like feature visualizations**, but this method appears to work **unconditionally**, whereas previous methods required label information.
- The results are **comparable to DeepDream-style methods**—perhaps not strikingly so to an untrained eye—but the approach comes with **nice theoretical justification**.
- However, direct quantitative comparisons to prior methods are lacking.

**Strengths:**

*(Note: my background is neither in NTK nor feature visualization; I read this paper from an outsider’s point of view.)*

- I like how the paper connects **NTK (specifically the Loss Tangent Kernel)** to feature visualization. Although some doubts remain (see Weaknesses), this connection feels **fundamental** and well-motivated within the rich NTK literature. Feature visualization has long relied on external regularizations to work; since NTK is an intrinsic property of the network, it provides a **plausible and principled justification** for defining such a distance metric.
- From a quick literature check, I tend to agree with the paper’s claim that this is the **first method demonstrating unconditional projection/generation**, which is a nontrivial achievement.
- The unconditional generation quality is **reasonable**, given that it’s a gradient-based visualization method with minimal generative priors.
- By casting the projection operation as a **score function**—hence the name *DSF*—the method conceptually aligns with **diffusion and flow models**, potentially enabling future extensions.
    - This connection is **ingenious**, allowing seamless integration with **conditional generation** (Sec. 5.2), including **CLIP-guided results**. These outputs, while not state of the art, look impressive.
    - Section 6.4 introduces sampling tricks that seem promising, though still underexplored.
- Overall, I genuinely **learned something new** from this paper—specifically, how NTK can be used as a distance measure to project arbitrary points onto a learned manifold.

**Weaknesses:**

- Since optimization appears to start from $\mathcal{N}(0, I)$, it’s unclear whether the proposed distance metric is **well-defined everywhere** consider that most discriminative models never train on noisy inputs and may behave in an arbitrary manner under such inputs. The paper doesn’t discuss this, and I think it deserves attention.
- Currently, an augmentation (at least flipping) is required to obtain the objective function (Eq 11). To make this method truly universal, we should also the case where the model isn’t trained with any augmentation. How can one justify using such augmentation in the objective function?
- The paper claims relevance to **explainable AI**, but the evidence provided is weak. While the visualizations are interesting, the claim isn’t substantiated in real-world scenarios. For example, one could test whether DSF helps uncover known dataset or model biases, and quantitatively measure how well it reveals them.
- From an untrained eye, the visual improvement over **DeepDream-style** results isn’t very clear. The comparison in Appendix Fig. 11 doesn’t use the same model/dataset across methods, making cross-method evaluation impossible.
- The visualization quality is still **inferior to methods using generative priors**—though this might not be a real drawback, as DSF deliberately avoids such priors and thus remains “truer” to the discriminative model.
- The paper fails to convey **intuition behind the DSF metric**—what does “close” or “far” mean under this metric? What kind of image changes correspond to those distances? A section probing the **properties of the DSF metric** would be helpful.
    - For instance, how does changing the initial noise condition affect the final output? Does colored (biased) noise lead to colored outputs?
    - How diverse are the outputs DSF can produce? Some **qualitative or t-SNE-like visualizations** would be helpful to show coverage and variability.
- Overall, the paper makes a **solid and original contribution**, but falls short in **discussion and interpretability**.

### **Minor Suggestions**

- Line 34: “*as we cannot claim to understand what we cannot create*.” — This isn’t a factual statement. Consider changing the tone or adding quotation marks.
- Include an **algorithm block** summarizing the full projection process to make replication easier.
- Clearly list **which augmentations** are used (is it only flipping?).

**Questions:**

- Is the distance metric introduced by the paper well-defined everywhere, especially when optimization starts from a random initialization (e.g., $\mathcal{N}(0, I)$)?
- How can the paper substantiate its claims about explainability or interpretability in real-world scenarios (e.g., detecting dataset/model biases) better?
- Can the paper include visualization comparisons on the same model/dataset for easy comparison against other methods?
- What is the intuition behind the DSF metric?
    - What does it mean for two points to be “close” or “far” under this metric?
    - What kind of mental picture should readers have when trying to understand DSF geometrically or perceptually?
    - How does changing the initial noise condition affect the final projected output?
- How diverse or mode-covering are the outputs that DSF can produce?
- What role do augmentations perform during optimization?
    - Is there a way to generalize this method to models that are not trained with augmentations?

---

> ### Author Response · Authors · 2025-11-21
> **Responses to Questions (1)**
>
> We sincerely thank the reviewer for the thoughtful and encouraging feedback. We are particularly grateful for your recognition of the fundamental connection between NTK and feature visualization, the achievement of unconditional generation, and the ingenious casting as a score function. Your comment that you "genuinely learned something new from this paper" is deeply appreciated and motivating.
>
> We have carefully addressed your concerns and questions below.
>
> ---
>
> ### Responses to Weaknesses and Questions
>
> **W1 & Q1. Is the distance metric well-defined everywhere, especially when starting from random noise?**
>
> Yes, the DSF metric is **well-defined everywhere in a technical sense**, though the distance values may be very large for inputs far from the manifold.
>
> **Why the Metric is Well-Defined:**
>
> The DSF metric (Eq. 10) measures the distance between the "hypothetical model that would result from training on the candidate sample" and the "actual pre-trained model." Crucially, this distance is computed in the **functional space defined by the Loss Tangent Kernel**, not directly in pixel space.
>
> For any input $x$ (including random Gaussian noise), we can compute:
> 1. The loss gradients $\nabla_\theta \mathcal{L}(f(x; \theta))$
> 2. The implied parameter update direction via the LTK
> 3. The distance between this implied model and the actual trained model
>
> This computation is always mathematically valid, regardless of whether $x$ resembles training data.
>
> **What Happens with Random Noise:**
>
> When we start from random Gaussian noise, the DSF metric will yield a **very large distance** because such noise would induce a hypothetical model drastically different from the trained model. However, this large distance value is still well-defined and meaningful: it correctly indicates that the noise is far from the data manifold.
>
> **How Optimization Works:**
>
> Gradient descent then reduces this distance by moving $x$ toward regions that would induce models closer to the trained one. As optimization progresses, the distance decreases and $x$ converges to the manifold. The fact that we empirically observe successful convergence from pure noise across diverse architectures (ResNet, ViT, DETR) confirms that the metric provides useful gradients even in these extreme regions of input space.
>
> ---

---

> > ### Author Response · Authors · 2025-11-21
> > **Responses to Questions (2)**
> >
> > **W5 & Q4. Intuition behind the DSF metric**
> >
> > **1. "A model is defined by its training data"**
> >
> > We view the training process as a mapping from a dataset to model parameters. DSF inverts this: it maps a candidate sample to the model it would produce if trained on it. The DSF metric thus measures the distance between the "model implied by the candidate sample" and the "actual pre-trained model."
> >
> > **2. Meaning of "Close" and "Far"**
> >
> > - **"Close"**: A sample is "close" if including it in the training set would not significantly change the current model parameters. The sample is consistent with the patterns, gradients, and statistics the model has already learned (i.e., it lies on the data manifold).
> >
> > - **"Far"**: A sample is "far" if it induces gradients that contradict or diverge from the current model state, implying the sample follows a distribution fundamentally different from the training data.
> >
> > **3. Geometric Mental Picture**
> >
> > Readers should visualize the process as "finding a dataset that reproduces the trained model." The pre-trained model is a fixed point in "Model Space." We optimize input pixels so that they map to this fixed point. If the mapping lands close to the pre-trained model, the input effectively "represents" the original training manifold.
> >
> > ---
> >
> > **Q5. How does changing the initial noise condition affect the final output?**
> >
> > Please see **Common Response C2**. In brief: the specific choice of initial noise has negligible impact on convergence or validity of generated images. Diversity is driven by the stochastic augmentation operator during optimization rather than initial conditions.
> >
> > ---
> >
> > **Q6. How diverse or mode-covering are the outputs?**
> >
> > We empirically validated that DSF produces diverse outputs covering various modes of the data manifold without signs of mode collapse.
> >
> > **Random Generation**: We present a grid of **400 randomly generated samples without any curation** in the Appendix. The results show a wide variety of object classes, poses, and backgrounds, confirming that the optimization landscape of DSF is rich and non-trivial.
> >
> > ---
> >
> > We believe these clarifications address your concerns about discussion and interpretability. Thank you again for your valuable feedback and for recognizing the fundamental nature of our contribution.

---

### Author Response · Authors · 2025-11-21
**[C1] Explainability & Bias Detection**

**Question:** How can DSF substantiate its claims about explainability and interpretability in real-world scenarios, such as detecting dataset/model biases?


We substantiate our claims by demonstrating how DSF visualizes the "inductive biases" and "data priors" inherent in the model's training objective through two complementary mechanisms:

**1. Visualizing Inductive Biases via Generated Artifacts:**

By observing the generated samples (Fig. 3), we can infer how the model processes data:
- **Object-Centric Bias:** Images generated from DETR (Object Detection) and ResNet (Classification) exhibit a strong bias towards centered, single objects with consistent scales, reflecting the nature of their training objectives.
- **Scale Invariance:** In contrast, samples from DINO (Self-Supervised Learning) show diverse scales and cluttered scenes, revealing that the self-supervised objective learns a more global, scale-invariant representation.

**2. Quantitative Analysis of Dataset Bias:**

By analyzing just 400 uncurated generated images, we can gain substantial intuition about the underlying data distribution. Examining the population of generated samples reveals inherent biases in the training data:
- **Animal Distribution:** Cats and dogs appear most frequently, while rare species are significantly underrepresented.
- **Architecture Distribution:** Christian architecture (churches) dominates, followed by parks and similar structures.

Since DINOv2's training dataset (LVD-142M) is primarily constructed by retrieval from ImageNet-22k and Google Landmarks [1], these patterns directly inherit the compositional biases of these source datasets—ImageNet-22k's heavy concentration of common domestic animals over rare species, and Google Landmarks' predominance of churches as the most frequent category, followed by parks and museums [2]. This observation suggests that the model's learned representations faithfully reflect—and potentially amplify—the categorical imbalances present in the curated seed datasets, particularly favoring common Western-centric visual concepts over underrepresented categories.

[1] Oquab et al., "DINOv2: Learning Robust Visual Features without Supervision," TMLR 2024.
[2] Weyand et al., "Google Landmarks Dataset v2," CVPR 2020.

---

### Author Response · Authors · 2025-11-21
**[C2] Initial Noise Sensitivity**

**Question:** How does the choice of initial noise $x_0$ affect output diversity? Is the generation process stable with respect to the initial noise distribution?


The specific choice of initial noise has **negligible impact** on convergence because DSF is a gradient-based optimization process that operates on a well-defined distance metric.

**Why Initial Noise Choice is Irrelevant:**

From Eq. (12), our generation process follows:
$$x_{t+1} = x_t - \eta_t \nabla_x d(\nabla_\theta f(x_t; \theta)\lambda, -\theta)$$

This is a standard gradient descent process where:
1. **Any starting point provides valid gradients**: Whether $x_0$ is Gaussian noise, uniform noise, or any other distribution, we can compute $\nabla_x d(\cdot)$ to obtain the descent direction toward the manifold.
2. **The manifold is the attractor**: The optimization objective (minimizing distance to the trained model in functional space) naturally pulls any initial point toward the data manifold, regardless of where we start.
3. **Local properties don't matter globally**: While different initializations start at different distances from the manifold, the gradient descent process ensures convergence to the manifold as long as the distance metric is well-behaved (which it is, as the LTK-based distance is continuous).

**Source of Diversity:**

The diversity in generated outputs comes not from initial noise, but from the **stochastic augmentation operator** $\mathcal{A}(x_t)$ in Eq. (11). During each optimization step, random augmentations (flips, crops, color jitter) are applied, and this stochasticity introduces variation in the trajectory. Even with identical initial noise, different random seeds for augmentations will produce different final outputs.

**Empirical Evidence:**

- **Starting from meaningful images** (Figure 5): When initialized with real images, the optimization preserves overall semantics while adjusting details, demonstrating that the process respects the initialization when it's already near the manifold.
- **Starting from pure noise**: Different random seeds lead to diverse outputs without mode collapse, confirming that the manifold contains rich structure that can be explored from various starting points.

In summary, initial noise simply determines the starting distance from the manifold, but the gradient-based optimization and augmentation stochasticity determine the final output, making the method robust to initialization.

---

### Author Response · Authors · 2025-11-21
**[C3] Quantitative Evaluation**

**Why Direct Comparison is Impossible:**

To demonstrate this infeasibility empirically, we evaluated existing discriminative inversion methods using Q-align [1], a unified image quality assessment metric. As shown in Table A1:

- **Discriminative Inversion Methods:** Existing methods (MTT: 0.00097, Energy-Based: 0.00376, Robust Classifier: 0.02398) produce outputs with near-zero quality scores, indicating they generate noise-like artifacts rather than recognizable images at high resolution ($256 \times 256$).

- **DSF (Ours):** Achieves a quality score of 0.0678, demonstrating that it successfully generates **recognizable, semantically meaningful images** from discriminative models—something no prior method has achieved at this scale.

**Position Relative to Generative Models:**

For reference, we also include dedicated generative models (StyleGAN: 0.4790, DDPM: 0.3389). While DSF does not match the quality of models explicitly trained for generation, this comparison is fundamentally different:
- Generative models are **trained** with generative objectives
- DSF is **zero-shot** from off-the-shelf discriminative models

**Our Contribution:**

The value of DSF lies not in outperforming dedicated generative models, but in **unlocking generative capabilities from discriminative models without any retraining**—a capability that existing discriminative inversion methods fail to provide at high resolution.

| Method | Architecture | Dataset | Paper | Quality Score |
|--------|-------------|---------|-------|---------------|
| **Discriminative Inversion Methods** |
| MTT | ResNet18 | TinyImageNet | Dataset Distillation (CVPR 2022) | 0.00097 |
| JEM | WideResNet 28-10 | CIFAR10 | Energy-Based (NeurIPS 2020) | 0.00376 |
| Robust Classifier | WideResNet 28-10 | CIFAR10 | Discriminative Generation (NeurIPS 2019) | 0.02398 |
| **DSF (Ours)** | ViT | LVD-142M | This work | **0.0678** |
| **Reference: Dedicated Generative Models** |
| StyleGAN | - | - | CVPR 2019 | 0.4790 |
| DDPM | - | - | NeurIPS 2020 | 0.3389 |

**Table A1:** Quantitative comparison using Q-align quality scores. The quality metric is derived from Q-align [1], which allows for a unified quality score to be measured across different datasets. Existing discriminative inversion methods (rows 1-3) produce near-zero quality scores, indicating noise-like outputs at high resolution. DSF achieves recognizable image generation from discriminative models (0.0678), while dedicated generative models (StyleGAN, DDPM) achieve higher scores as expected for models explicitly trained for generation. Visual samples are shown in Appendix.

[1] Wu et al., "Q-Align: Teaching LMMs for Visual Scoring via Discrete Text-Defined Levels," ICML 2024.

---

### Author Response · Authors · 2025-11-21
**[C4] Computational Complexity**

**Question:** How computationally expensive is the DSF generation procedure? How does it compare to standard diffusion sampling?

**Response:**

The generation process is **computationally efficient and practical**, with costs comparable to standard diffusion models.

**Specific Metrics:**
- **Wall-Clock Time:** Generating a single $$256 \times 256$$ image takes approximately **36 seconds** on a single NVIDIA RTX A6000 GPU.
- **Iterations:** We typically use $T \approx 200$ iterations per image.
- **Comparison:** This is comparable to the inference time of standard diffusion models with similar sampling steps, confirming that our method is viable for research and application.

**Per-Step Complexity:**

One iteration of DSF involves a forward and backward pass through the network, making it roughly $2\times$ the cost of a standard inference step. The theoretical cost is $O(D \times P)$, where $D$ is the dimension of the generative candidate and $P$ is the parameter size.

**Optimization for Efficiency:**

Crucially, we do not necessarily need to compute gradients for all parameters. For deep models like DINOv2 (ViT), we found that backpropagating through only the **initial subset of layers** (e.g., the last few transformer blocks) is sufficient to obtain meaningful gradients for the input pixels. This "partial gradient" approach significantly reduces memory and computation costs without sacrificing generation quality.

---

### Author Response · Authors · 2025-12-03
**Final Remark for Area Chair**

To assist your assessment, we provide this summary outlining our core contributions with reviewer consensus, followed by how we addressed all major concerns during the discussion period.

---

### 1. Core Contributions & Reviewer Consensus

All four reviewers acknowledged the core contributions of our work, despite differences in final ratings.

**First Unconditional Generation from Discriminative Models.** We introduce the Discriminative Score Function (DSF), enabling unconditional image generation from any off-the-shelf discriminative model without modification or retraining. Reviewers noted this as "a nontrivial achievement" [YWkD], with "novel conceptual framework" [SzrC] and "novel method with advantages over existing methods" [tjrt]. [Acknowledged: All four reviewers]

**Architecture & Task Agnostic Framework.** DSF operates across diverse architectures (ResNet, ViT, DETR) and training objectives (classification, detection, self-supervised learning including DINO and CLIP). Reviewers described this as "architecture-agnostic and requires no retraining, an elegant practical feature" [SzrC], "simple to implement" [EMNs], and demonstrating "ability to scale to large-scale models" [tjrt]. [Acknowledged: SzrC, EMNs, tjrt]

**Novel Theoretical Connection.** We bridge NTK theory with generative modeling by using the Loss Tangent Kernel to define a score function analog. Reviewer YWkD stated: "I genuinely learned something new from this paper, specifically, how NTK can be used as a distance measure to project arbitrary points onto a learned manifold." [Acknowledged: YWkD, SzrC]

**Interpretability & XAI Applications.** DSF reveals model biases and data entanglements through generation, serving as an interpretability tool for vision models. [Acknowledged: SzrC, YWkD]

---

### 2. Addressed Concerns

We conducted extensive new experiments and analysis to address all major concerns raised during review.

**A. Quantitative Evaluation [EMNs, SzrC, tjrt].** We provided Q-align quality scores (Common Response C3). While existing discriminative inversion methods produce near-zero scores (MTT: 0.00097, JEM: 0.00376, Robust Classifier: 0.02398), DSF achieves 0.0678, demonstrating successful high-resolution generation where prior methods fail.

**B. Computational Complexity [EMNs, SzrC].** Generation takes ~36 seconds per image on A6000 (Common Response C4). Partial gradient computation through subset of layers reduces cost without sacrificing quality.

**C. Initial Noise Sensitivity [YWkD, EMNs].** We demonstrated robustness to initialization choice (Common Response C2). Diversity arises from stochastic augmentation during optimization, not initial noise distribution.

**D. Explainability Claims [YWkD].** We substantiated XAI claims by analyzing 400 uncurated generated samples (Common Response C1), revealing dataset biases (cats/dogs dominance from ImageNet-22k, church architecture prevalence from Google Landmarks) that trace directly to DINOv2's training data composition.

**E. DSF Metric Intuition [YWkD].** We provided geometric interpretation: DSF measures distance between "the model implied by a candidate samples" and "the actual pre-trained model" in functional space. "Close" means the samples are consistent with learned patterns; "far" means it would induce contradicting gradients.

**F. Writing Quality [tjrt, EMNs].** We completely revised the introduction and corrected all grammatical and formatting issues. Reviewer tjrt, who initially rated 2 primarily due to writing concerns, explicitly confirmed after reviewing our revisions: "The flow of the first paragraph is significantly better" and raised their score to 6.

---

### 3. Summary

| Concern | Raised By | Addressed In | Key Result |
|---------|-----------|--------------|------------|
| Quantitative evaluation | EMNs, SzrC, tjrt | C3 | Q-align: 0.0678 vs prior methods <0.024 |
| Computational cost | EMNs, SzrC, tjrt | C4 | ~36s/image|
| Initial noise sensitivity | YWkD, EMNs, SzrC | C2 | Robust; diversity from augmentation |
| Explainability claims | YWkD, tjrt | C1 | Dataset bias detection validated |
| DSF metric intuition | YWkD | Reply | Geometric interpretation provided |
| Writing quality | tjrt, EMNs | Revision | tjrt score increased 2→6 |

---

---

> ### Author Response · Authors · 2025-12-03
> **Final Assessment**
>
> Three reviewers reached positive conclusions after the discussion period:
>
> - **SzrC (8)**: Maintained strong support, explicitly stating in their response: "I support its acceptance given its potential to inspire further work in unified discriminative-generative modeling."
>
> - **YWkD (6)**: Provided highly positive evaluation in their review: "I genuinely learned something new from this paper, specifically, how NTK can be used as a distance measure to project arbitrary points onto a learned manifold." They also noted the contribution as "fundamental and well-motivated" and the unconditional generation as "a nontrivial achievement."
>
> - **tjrt (6)**: Initially rated 2 due to writing concerns, but after reviewing our revisions, confirmed "The flow of the first paragraph is significantly better" and raised their score to 6.
>
> Reviewer **EMNs (4)** did not respond during the discussion period. Their primary concerns, writing quality and quantitative evaluation, are identical to tjrt's initial concerns, which were demonstrably resolved as evidenced by tjrt's score increase from 2 to 6.
>
> We respectfully request the AC consider that all substantive concerns have been addressed, and the remaining below-threshold score reflects unacknowledged revisions rather than unresolved issues.

---

### Meta-Review · Area_Chair_MyCH · 2025-12-31

**Summary:**

In this paper, the reviewers are confident in the paper’s ability to generate using an already trained discriminator. This paper uses the discriminator to get closer to the manifold from which the data was generated by using the gradient, similarly to adversarial training, but in the opposite direction.

The paper's image generation and editing quality are minimal compared to state-of-the-art methods, so its value lies in its innovative use of an already trained discriminator to generate plausible images.  The issue with this solution is that there is no path forward to improve the generator because the manifold is always poorly described by discriminators, and their goal is to discriminate, so it will be hard for this paper to be more than a curiosity.

The paper also has several limitations:

1 A discriminative model predicts p(y|x), so it does not model p(x) in any way. In contrast, a generative model predicts p(y, x), so it is easy to turn a generative model into a discriminative model. Still, it should not be possible to do it the other way around. However, there is ample experience using semi-supervised learning to improve discriminative models, especially for anti-causal relations, where knowing p(x) without labels helps discrimination, indicating the need to learn part of the x distribution to perform well. Even when four semi-supervised papers are cited, the semi-supervised concept is not named in the paper.
2 The authors fail to recognise that the GPT model for testing is a discriminative model that can generate. This is the whole point of the OpenAI papers from 2018 and 2019, which show that these models achieve zero-shot learning when trained with sufficient data. To be clear, GPT models are trained to predict the next word/token. From that perspective, they are taught as a discriminative model. When they use their own predictions to generate, they become a generative model that was not trained for it.
3 The next step of this paper is already what we know of using diffusion models to generate images, which are models trained to discriminate and then are used to generate images.
4 Finally, the connection with adversarial training should be further emphasised, as it is doing the adversarial examples but in the opposite direction, going towards the manifold instead of away from it.

All of these topics could not be discussed with the reviewers given the issue of the leakage of their names, but I would have raised those with the reviewers during the discussion period.

**Reviewer Concerns:**

The reviewers' concerns were minimal and addressed mainly by the authors. However, as I show in my meta-review (new review), the paper has many limitations and should not be accepted for publication.

**Reviewer Scores:**

They could have raised them, except that the AC had intervened, as noted in my comments above.

This paper could have been accepted, but for my intervention. You might consider that if you want the AC to only call balls and strikes,

---

### Decision · Program_Chairs · 2026-01-26

Reject